# Born a Transformer – Always a Transformer? On the Effect of Pretraining on Architectural Abilities

**Mayank Jobanputra[1]\*, Yana Veitsman[1]\*, Yash Sarrof[1], Aleksandra Bakalova[1],**
**Vera Demberg[1], Ellie Pavlick[2], Michael Hahn[1]**

[1]Saarland University    [2]Brown University

{mayank,mhahn}@lst.uni-saarland.de

## Abstract

Transformers have theoretical limitations in modeling certain sequence-to-sequence tasks, yet it remains largely unclear if these limitations play a role in large-scale pretrained LLMs, or whether LLMs might effectively overcome these constraints in practice due to the scale of both the models themselves and their pretraining data. We explore how these architectural constraints manifest after pretraining, by studying a family of *retrieval* and *copying* tasks inspired by Liu et al. [2024a]. We use a recently proposed framework for studying length generalization [Huang et al., 2025] to provide guarantees for each of our settings. Empirically, we observe an *induction-versus-anti-induction asymmetry*, where pretrained models are better at retrieving tokens to the right (induction) rather than the left (anti-induction) of a query token. This asymmetry disappears upon targeted fine-tuning if length-generalization is guaranteed by theory. Mechanistic analysis reveals that this asymmetry is connected to the differences in the strength of induction versus anti-induction circuits within pretrained transformers. We validate our findings through practical experiments on real-world tasks demonstrating reliability risks. Our results highlight that pretraining selectively enhances certain transformer capabilities, but does not overcome fundamental length-generalization limits[1].

## 1 Introduction

Transformers [Vaswani et al., 2017] have become the backbone of most LLMs [Radford et al., 2019, Brown et al., 2020, Touvron et al., 2023]. Given the widespread usage of LLMs, there has been a long line of research to understand their theoretical underpinnings. Early analyses showed that transformers can approximate any sequence-to-sequence function at fixed input length [Yun et al., 2019] and even simulate Turing machines [Pérez et al., 2019] when given unbounded compute. However, subsequent studies have uncovered fundamental expressibility and learnability limitations, finding classes of problems which transformers cannot express across arbitrary input lengths [e.g. Hahn, 2020, Strobl et al., 2024, Merrill and Sabharwal, 2023, Sanford et al., 2024], or which transformers systematically struggle to learn and generalize on [e.g. Hahn and Rofin, 2024, Zhou et al., 2024]. Empirical work confirms that small transformers trained from scratch do instantiate such limitations [e.g. Bhattamishra et al., 2020, Huang et al., 2025, Delétang et al., 2023, Zhou et al., 2024]. For instance, transformers trained from scratch struggle to generalize to longer or more challenging examples even on tasks as simple as copying a long string with repeated symbols [Zhou et al., 2024], or retrieving which token followed the last occurrence of some query token in a long context [Liu et al., 2024a] — findings in close agreement with theoretical arguments about transformer's learning biases [Huang et al., 2025].

---

\*These authors contributed equally.

[1]https://github.com/lacoco-lab/always_a_transformer

However, it remains unclear if such results on expressiveness and generalization have any bearing on large-scale pretrained models. Modern LLMs have many more layers, and many orders of magnitude more parameters than the small-scale transformers typically used for testing theoretical limitations. Model scale alone might make theoretical limitations irrelevant at realistic input lengths. Perhaps even more importantly, LLMs are pretrained on a massive scale, with trillions of tokens of varied data that may impart fundamentally new inductive biases and algorithmic abilities [Rytting and Wingate, 2021] that may be invisible in models trained from scratch on the target task alone [Amos et al., 2024]. Indeed, LLMs' capabilities scale in ways that are difficult to foresee: empirical "laws" relate performance to model, data, and compute budget [Kaplan et al., 2020, Hoffmann et al., 2022] and qualitatively new behaviors such as in-context learning, chain-of-thought reasoning, and tool use emerge abruptly at scale [Wei et al., 2022, Ganguli et al., 2022, Bubeck et al., 2023]. Since pretraining reshapes both the model's inductive biases and the sub-circuits it relies on, length-generalization failures observed in small models might not transfer automatically to their large-scale counterparts.

**Scope.** We focus on *retrieval* and *copying* as fundamental operations that provide a test bed for answering these questions. These operations are critical building blocks for many real-world LLM applications, such as question answering and summarization [Wiegreffe et al., 2025, Fan et al., 2024]. On a mechanistic level, retrieving from context via induction heads is key to performing in-context learning [Olsson et al., 2022, Song et al., 2025, Crosbie and Shutova, 2025], while faithfully copying exact spans makes retrieval-augmented generation more effective [Yu et al., 2024]. Understanding the potential impact of architectural limitations on LLMs is thus of great interest for these tasks.

In this work, we revisit the theory of length generalization for these operations, ask how its predictions hold up for pretrained transformers, and diagnose which abilities are amplified by pretraining and which limitations persist. We study a family of retrieval and copying tasks[2] (Figure 1) inspired by Liu et al. [2024a], Zhou et al. [2024]. We organize retrieval into two umbrella categories of *unique* and *non-unique* induction, respectively, in which the query token appears exactly once or multiple times respectively. In each variant, the model must return a token located either *before* or *after* a designated occurrence of that query token. We make an analogous distinction in copying between copying unique tokens or repeated tokens, and between copying in the forward or the reverse direction.

All retrieval and copying tasks we consider are expressible by transformers in principle; however, theory predicts systematic differences in length generalizability[3]. Specifically, we show (Section 3) that theory predicts transformers to have a *uniqueness bias* (i.e., unique-token induction is easier than its non-unique counterpart) [Huang et al., 2025, Zhou et al., 2024]; but for it to not have a *directional bias* (i.e., that forward and reverse variants are equally hard). Theory also predicts that within non-unique induction, retrieving is easier from the first occurrence of the query character than from the last one.

However, given the impressive capabilities of transformer-based LLMs, it is reasonable to posit that the massive scale of both such models and of pretraining might overcome these limitations. For instance, high prevalence of non-unique copying in natural language might lead transformers to learn specialized circuits or components for handling such situations. Here, we directly test whether these architectural limitations remain relevant to pretrained LLMs. We study 8 pretrained LLMs from Llama-3 and Qwen2.5 families [Grattafiori et al., 2024, Yang et al., 2024]. Our findings show that large-scale pretraining produces a directional bias (Section 4.1). In addition to showing our results on synthetic settings, we show how these limits might surface and the reliability risks they pose for practitioners in more natural settings as well (Section 4.2). While targeted fine-tuning can even out the directional bias born out of pretraining, we find that the fundamental architectural uniqueness bias still binds pretrained transformers (Section 4.3).

## 2 Background, Notation & Definitions

**Task Description.** We examine two primary task types, **Retrieval** and **Copying**, each instantiated under *Unique* and *Non-unique* conditions, as illustrated in Figure 1. In **Retrieval**, the sequence has the form $X = \langle bos \rangle x_1, \ldots x_{N-1} \langle sep \rangle x_N$, where each $x_i \in$ some alphabet $\Sigma$ and $x_N = q$ is the query token. The query $q$ appears $t$ times in the context, indexed as $q_1, \ldots q_t$ with $1 \leq q_1 < \cdots < q_t \leq N$. Transformer has to predict a single token as the output given the input. For the *Unique* condition, we

---

[2]i.e., copying of the *input*, not the *training data*, similar to Jelassi et al. [2024]

[3]ability to correctly solve tasks on instances longer than those seen during training

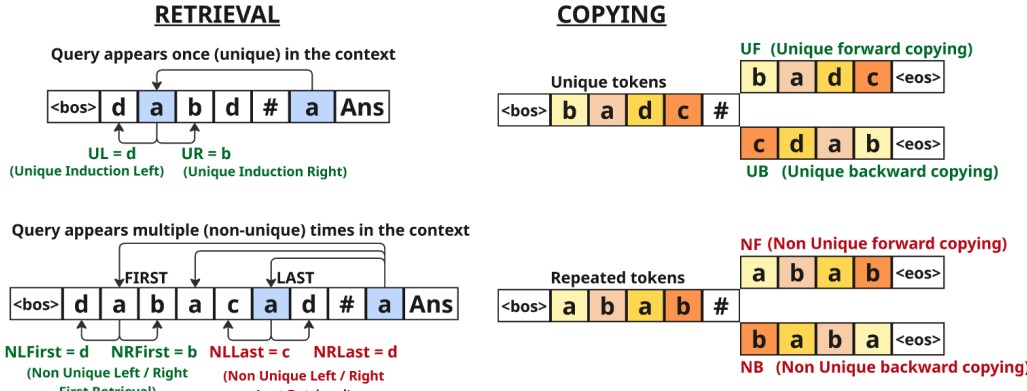

Figure 1: Overview of our task variants (formal definitions in Section 2). **Retrieval:** A '#' marks the separator; the token that follows is the query that may appear once (unique) or multiple times (non-unique). With non-unique queries we return the token immediately left/right of either the first or the last occurrence, creating 6 sub-tasks in total. **Copying:** We want to model to copy the context which consists of either only unique tokens or repeated tokens in the forward or reverse direction. Tasks in green lie in C-RASP[pos] and length-generalize; those in red do not (proofs in Section 3).

define **UL** (Unique Induction Left) and **UR** (Unique Induction Right) respectively when the output expected is the token preceding or following the query's single occurrence, i.e., $x_{q_1-1}$ and $x_{q_1+1}$. Under the *Non-unique* condition, **NLFirst** (Non-unique Left First) and **NRFirst** (Non-unique Right First) use the first occurrence to predict $x_{q_1-1}$ and $x_{q_1+1}$ as the output, while **NLLast** (Non-unique Left Last) and **NRLast**[4] (Non-unique Right Last) use the last occurrence to produce $x_{q_t-1}$ and $x_{q_t+1}$ respectively.

For **Copying**, the sequence is $X = \langle bos \rangle x_1 \ldots x_N \langle sep \rangle$ and the expected output is a continuation $x_{N+1} \ldots x_{2N}$. The decoder is expected to replicate the latter segment based on the first. For *Unique* conditions all of the tokens in $X$ are unique. **UF** (Unique Forward copying) and **UB** (Unique Backwards copying) denote forward and reverse copying of the input segment. For *Non-unique* settings there is no uniqueness constraint on the $x_i$ values and **NF** and **NB** also denote the forward and reverse copying of the input segment.

**Induction and Anti-Induction Heads.** Induction heads were characterized by Elhage et al. [2021] as part of a two-head circuit present in autoregressive transformers and were later shown to be critical to performing in-context learning [Olsson et al., 2022].A **previous-token head** in layer $\ell$ writes the embedding of the immediately *preceding* token into the residual stream. An **induction head** in some layer $\ell' > \ell$ then queries for positions whose stored "previous token" equals the current token and copies the corresponding value vector forward, enabling the model to predict the token *after* a

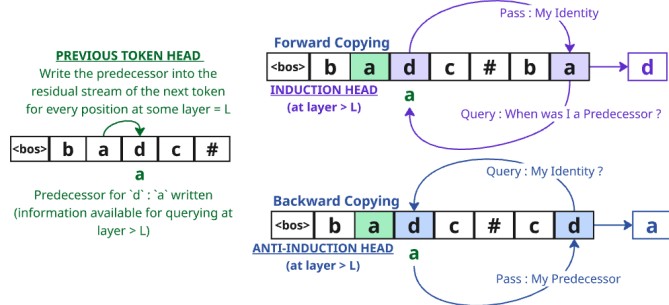

Figure 2: Illustration of the induction and anti-induction Circuits.

repeated bigram. Our UR retrieval as well as the UF copying task setups are not novel and are meant to test this induction circuit ability and compare with our other task setups.

We hypothesize that, by symmetry, an **anti-induction circuit** should in principle exist and be able to copy information *backwards*. Here, the same previous-token head caches the token immediately to the left, but is paired with an **anti-induction head** that *looks back* to the earlier occurrence of the *same*

---

[4]The NRLast variant is conceptually equivalent to the Flip-Flop task of Liu et al. [2024a].

token. Because that earlier token received the value of its own predecessor from the previous-token head, the anti-induction head can retrieve that predecessor, allowing the model to predict one step to the *left*. Such a two-head motif should thus enable exact solutions to our UL and UB tasks. *Retrieval heads* of Wu et al. [2025] subsume both induction and anti-induction heads (see Appendix E for discussion).

## 3 Theoretical Length Generalization Guarantees for Our Tasks

We derive theoretical predictions for transformers' length-generalization abilities on the retrieval and copying tasks introduced above, using the framework of Huang et al. [2025], which links length-generalizability to expressiveness in C-RASP [Yang and Chiang, 2024]. Of particular interest is **C-RASP** [Yang and Chiang, 2024], which restricts the usage of positional information and arithmetic operations. Huang et al. [2025] further added positionally-aware primitives to C-RASP to construct the language C-RASP[pos][5]. Doing so, they established–both theoretically and empirically–a tight link between the existence of a C-RASP[pos] program for a task and the ability of *decoder-only* transformers (with absolute or no positional encodings) to length-generalize on that same task: in particular, if C-RASP[pos] program exists, a decoder-only model provably generalizes to longer inputs under a specific formal model of training (Theorem 7 in Huang et al. [2025]). More precisely, this guarantee applies to APE Transformers under an idealized training procedure, where all training data up to some maximum input length $N$ are available and the exact minimizer of a regularized loss is found. Under these specific conditions, the existence of a C-RASP[pos] program for that task guarantees length generalization to larger lengths. While theory has only been worked out for this specific setup, empirically we observe converging results across different PE types and realistic SGD-based training (as shown in Huang et al. [2025] as well as in the experiments in Appendix B). As a converse to this length generalization guarantee, extensive experimental evidence in Huang et al. [2025] suggests that, if no C-RASP program exists, such models *fail* to generalize to longer inputs.[6]

We therefore adopt the following strategy for providing length generalizing guarantees across our retrieval and copying tasks:

*Construct a* C-RASP[pos] *program* $\implies$ the task length-generalizes to arbitrary $n$.

We start by considering the "Right" versions of our Retrieval tasks. In the unique case (UR), a transformer can solve retrieval using a standard induction circuit (Figure 2): an induction head retrieves the symbol that followed the unique occurrence of the query. These operations can be simulated in C-RASP[pos] (Lemma 4), ensuring a theoretical length generalization guarantee. In the non-unique case, expressibility – and hence the length generalization guarantee – varies:

**Theorem 1.** *NRFirst is expressible in* C-RASP[pos]. *NRLast is not expressible in* C-RASP[pos].

The proof is in the Appendix (Lemmas 5 and 6). An extension of the induction circuit performs NRFirst: first, at each $x_i$, an attention head checks whether this symbol has appeared previously. Given a query $w$, an induction circuit then retrieves the symbol immediately following the (unique, if any exist) first appearance of $w$. A C-RASP[pos] program formalizes this (Lemma 5). On the other hand, the inexpressibility of NRLast is proven by reduction to the regular language $(a|b|e)^*ae^*$, provably not in C-RASP (Lemma 6). Regarding copying tasks, Huang et al. [2025] shows that UF is in C-RASP[pos], while NF is not. The construction for UF again uses an induction circuit.

So far, we have considered "right" and "forward" versions—in fact, in all tasks, C-RASP[pos] expressibility is equivalent for the left (backward) settings:

**Theorem 2.** *Across all tasks, there is no* C-RASP[pos] *expressibility difference between R vs L versions, and F vs B versions.*

---

[5]Referred to as C-RASP[local, periodic] in Huang et al. [2025]. See Appendix A.1.

[6]While Huang et al. [2025] focuses on absolute (APE) or no positional encodings (NoPE), our empirical results show that the conclusions also match the behavior of Rotary Positional encodings (RoPE) (See Appendix B). Full theoretical understanding for RoPE length generalization remains an interesting problem. One interesting aspect is that, while training RoPE for UF and UB did not show perfect generalization, we did obtain perfect generalization when instead autoregressively decoding from models trained for UR/UL. In APE, we obtained perfect generalization when training directly for UF and UB, in line with C-RASP[pos] expressivity.

That is, C-RASP[pos] expressivity is the same for UL and UR, the same for UF and UB, etc. (Corollary 9). All tasks are annotated in Figure 1 with their C-RASP[pos] expressibility. Experiments in Appendix B show that transformers trained from scratch on these tasks exhibit length generalization as predicted by the theoretical results.

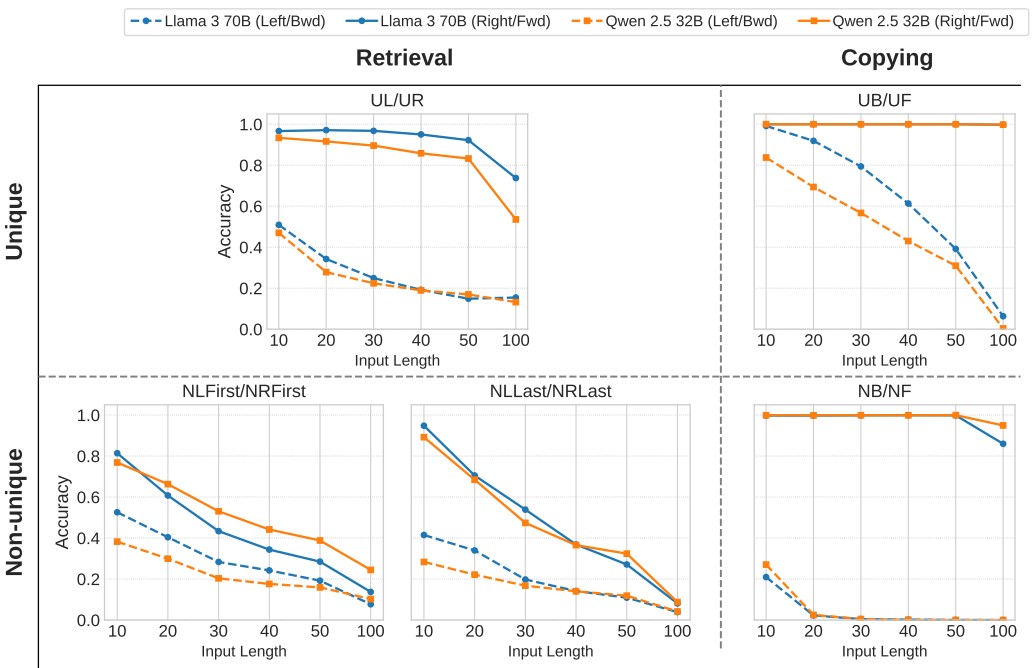

Figure 3: In-context accuracy for `Llama-3 70B` and `Qwen2.5-32B` across all our tasks averaged over three seeds. Across all settings, lengths, model size, and task type, we observe a Directional Bias: Retrieving the token to the left of the query token is always more difficult compared to the one to the right, provided all other things are constant. Similarly, copying in the forward direction is easier than copying backwards. Detailed prompts, similar performance graphs on other models (including instruction-tuned variants) are in Appendix C.

## 4   Experiments

### 4.1   Eliciting Abilities via In-Context Learning

We evaluate **pretrained completion** as well as **instruction-tuned** variants of the following models: `Llama3.1-8B`, `Llama3.1-70B`, `Qwen2.5-7B`, and `Qwen2.5-32B`. For retrieval as well as copying, each model is evaluated across various prompt variations. The prompt variations relate to the few-shot example size (examples are equal in length to the test string or are smaller but clearer) and the instruction templates (details on prompt templates in the Appendix C.1) with the templates adjusted for completion as well as instruction-tuned models. All experiments use 1,500 test strings for each length $10, 20, 30, 40, 50, 100$ generated under three independent seeds. We cap the input string length at 100 tokens to maintain a bounded vocabulary comparable to the unique setups; see Appendix C.4 for results at other lengths. The test strings contain spaced letters (including diacritics) and digits, to ensure that every character gets tokenized as an independent token and thus, the required uniqueness / repetitions can be maintained within the context, whenever required. Few-shot prompts contain $k = 5$ demonstrations, reserving one slot whenever an explicit explanation is required by the template. For each task that belongs to the same group i.e. for Unique (UR, UL) and for Non-Unique (NRLAST, NRFIRST, NLLAST, NLFIRST), we have the same strings on which we test the in-context ability with different expected answers to avoid the confound of different datapoints.

**Directional Bias: Pretraining induces a left-right asymmetry.** We found a persistent *directional bias* in all of the pretrained models we tested (Figure 3). For each task, retrieval or copying, the models consistently performs better on rightward tasks (UR, NRFirst, NRLast, UF, NF), across models,

prompt templates, and sequence lengths. The drop in performance for the leftward tasks cannot be attributed to lexical cues such as the words "right" or "forward" (explored in earlier work [He et al., 2025]) because the performance asymmetry persists even when we strip all natural-language instructions and provide only input–output pairs for the model to infer the task in-context. C-RASP[pos] predicts symmetric performance in transformers trained from scratch within a task family (Theorem 2). We further find that models trained from scratch do not exhibit such a left-right asymmetry, confirming it is not inherent to the architecture (Appendix B). We therefore hypothesize that pretraining instills this bias because induction is a more natural task than anti-induction and thus more induction head circuits get formed during pretraining as opposed to anti-induction circuits; Sections 4.3 and 4.4 probe this explanation in more detail.

**Uniqueness Bias: Unique tasks are easier to elicit than non-unique tasks.** Unique retrieval and copying setups both perform relatively well in rightward / forward direction. In contrast, in the non-unique settings, eliciting correct retrieval in any direction is challenging across task setups. We observe an interesting difference to what is expected for transformers trained from scratch: First, the underperformance of NRFirst in pretrained models does not align with what is expected for transformers trained from scratch (Theorem 1). Copying, by contrast, is easier to elicit than retrieval, even though copying requires predicting many tokens and hence offers more opportunities for error. Remarkably, Non-unique-forward (NF) remains near perfect throughout. As this experiment only considers input lengths up to 100 for comparable results vis-à-vis the unique settings, an interesting question is how copying performs at longer lengths. The next section therefore stress-tests in-context copying on longer paragraphs, seeking to reveal if the theoretically predicted difficulty of non-unique copying (NF) appears in realistic settings.

> **Takeaway**: Pretraining introduces a Directional Bias favoring retrieving and copying in the forward/right direction over the backwards/left. Pretraining also selectively amplifies the abilities of a transformer, as evident from the near perfect performance on copying repeated tokens, and poor performance on certain retrieval tasks, even though the latter are better learnable in a length-generalizable way for the architecture.

## 4.2 Testing Uniqueness Bias and Directional Bias in Natural Settings

**Testing Uniqueness Bias by Copying Lorem Ipsum Paragraphs.** In Section 4.1, forward copying showed near-perfect performance irrespective of uniqueness (UF and NF). We now ask whether the theoretically predicted asymmetry between unique and non-unique copying still impacts LLM behavior, by having models reproduce longer, naturalistic but semantically neutral text. We here do not focus on evaluating exact copying accuracy, instead focusing on understanding the nature of LLM mistakes.

While entirely incorrect outputs are easily detectable, subtle differences in nearly accurate copies represent the most challenging errors for users to identify. Therefore, we focus exclusively on the cases where the generated output is within 75% of the length of the expected output, targeting scenarios where the model clearly understands the copying task and produces plausible copies of the input. To confirm that our prompting setup is valid, we evaluate the models on copying exact spans from recent arXiv papers (April 2025), finding perfect performance.

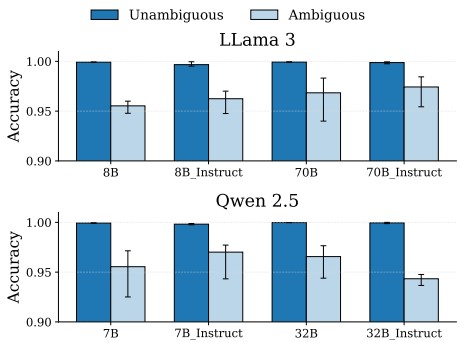

Figure 4: Failures in accurate copying of Lorem Ipsum paragraphs are associated primarily with ambiguous transition indices.

We design our copying benchmark to suppress semantic cues that could help the model break ambiguous copy chains. Specifically, each input consists of a randomized Lorem-Ipsum–style paragraph. Such content is unlikely to have appeared verbatim in the training corpus. Each paragraph has close to 500 tokens. See Appendix C.5.1 for results at longer lengths (upto 5k tokens).

We classify each token in the input as either *unambiguous* (uniquely predictive of the subsequent token) or *ambiguous* (potentially followed by multiple subsequent tokens and therefore is not perfectly

predictive of the subsequent token). We perform token-level alignments using the Needleman-Wunsch algorithm [Needleman and Wunsch, 1970], to detect token-level insertions, deletions, and substitutions. What we are left with is an alignment sequence consisting of 4 kinds of entries: *match*, *insert*, *delete* and *substitute*. We group consecutive entries in this alignment list if they are of the same type and call them *aligned spans* if the entries consist of only 'match' and *misaligned spans* if the entries grouped consist of any of the other types. We refer to indices where an aligned span finishes and a misaligned span starts as *transition indices*. These transition indices mark critical points where an error in copying started propagating during autoregressive generation. It should be noted that, even if there are just a few such misaligned spans, they can each be very long, especially for the ones consisting exclusively of 'insert'. Such hallucinated insertions can *at times* snowball into a long error chain from where the model never reorients itself to produce a correct continuation (Examples in Appendix C.5).

We categorize the transition indices as *unambiguous* or *ambiguous* based on the nature of the token at that index. Since an induction head would suffice to correctly predict the subsequent token following an unambiguous token, we hypothesized that transition indices would never be unambiguous. Conversely, ambiguous tokens were expected to be the source of where all of the *misaligned error span* began, as it has similarities to the failure modes expected in NF (Non-unique forward copying). Empirical results (Figure 4) confirm this hypothesis. Across all model variants tested (sizes, families, completion as well instruction tuned models), a consistent pattern emerges, the transition indices almost unilaterally always are ambiguous and cause the 'glitches' we see in the final copied output. While these glitches appear across models, their exact nature appears to be random. We did not find any pattern except for the fact that they were tied to ambiguous tokens.

**Testing Directional Bias in Git Commit History: Direction-Aware Copying.** While the Lorem Ipsum setup shows how glitches in forward copying might cause reliability issues, we now show how failures in backward copying impact real world tasks. We use a compact Git history manipulation task to exemplify when the right-over-left asymmetry seen in Section 4.1 surfaces in practice. Given a truncated `git log` (newest→oldest commit hashes), the model must list commits that need *reverting* – `git revert`(forward order) or must be *cherry-picked* – `git cherry-pick`(reverse order) onto a release branch.

Llama-3 models reproduce the forward list almost perfectly but accuracy drops when the order flips (Figure 5) – mirroring the asymmetry seen in the synthetic experiments (Section 4.1). Qwen models are not able to do the task at all, hence we exclude them from this analysis. The reversal presented here does not happen here at the token level, instead at the commit order level and is thus in principle more similar to a generalized copying setup. However, the theoretical results and expectations transfer to the generalized scheme (See Appendix for a detailed discussion A.2.1).

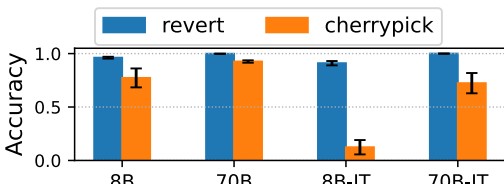

Figure 5: Git Commit History Manipulation also has the forward vs backward asymmetry seen in Section 4.1.

> **Takeaway**: Even though pretraining selectively amplifies certain abilities of the transformer, fundamental limitations inherent to the architecture have a marked impact even after the pretraining and need to be kept in mind while deploying these models in real-world tasks.

## 4.3 Fine-tuning Eliminates the Directional Bias, But Not the Uniqueness Bias

Our experiments have uncovered three failure modes: *(i)* a Directional Bias in pretrained models, *(ii)* poor performance on certain retrieval tasks that are *provably* length-generalizable, and *(iii)* the Uniqueness Bias: sporadic "glitches" on copying tasks that are *not* length-generalizable. Each of these could stem either from architectural limitations or from inductive biases imparted during pretraining. To disentangle the two, we carry out task-specific supervised fine-tuning and see which failures are removed. Since we wish to know whether the model has truly learned a *length-generalizable* solution, we evaluate on test sequences twice as long as those seen in training, following Huang et al. [2025]. A fully correct model should therefore achieve 100% accuracy on this test set; even a single error

then flags a reasoning flaw that could undermine reliability in real-world applications, an evaluation scheme similar to that of Liu et al. [2024a].

**Experimental Setup.** We fine-tune a pretrained language model —GPT-2 1.5B (APE)[7]— on each retrieval task plus unique forward and backward copying. Fine-tuning uses the standard next-token objective but masks the loss to the answer span only, mimicking inference, i.e., for retrieval tasks, just the answer token, and, for the copying tasks, all tokens after the EOS delimiter in the input, indicating the end of the input sequence. To ensure that APE models can generalize on longer positions than those seen during training, we apply the *positional offset* trick of Huang et al. [2025]: for every training input a random constant $\Delta \in [0, 200 - \ell_{min}]$ is added to all position indices, ensuring that all the positional embeddings are trained. All models see training examples whose input lengths are sampled uniformly from $[\ell_{\min}, 100]$, where $\ell_{\min}$ is the length of the shortest well-formed instance of the task. We confirmed that, prior to this fine-tuning, the base model's performance on each task was effectively zero. Details of hyperparameters and datasets are in Appendix D.

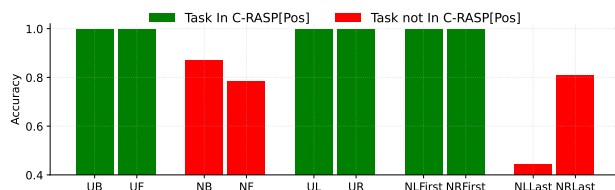

Figure 6: On the out-of-distribution dataset, tasks belonging to C-RASP[POS] (UL, UR, UF, UB, NLFIRST, NRFIRST) retain perfect accuracy, whereas NLLAST, NRLAST, NF, and NB, which are provably not in C-RASP[POS], suffer a drop in performance. The Directional Bias within task setups is largely absent from the fine-tuned models.

Our testbed setup for all of our tasks closely follows the setup of Bhattamishra et al. [2020], Huang et al. [2025], with an in-distribution bin (i.e., lengths $[\ell_{min}, 100]$) and an out-of-distribution evaluation bin (i.e., lengths $[101, 200]$). Similar to their setup, we report accuracy on our test sets only on seeds where training loss converges to 0. For our out-of-distribution evaluation bin on Retrieval tasks, we enforce that all instances of the query token $q$ appear exclusively in the *first half* of the context $\mathcal{C}$. This way distance between the relevant tokens to which the model needs to attend increases, thereby rigorously testing the model's ability to generalize to longer contexts. In such settings, superficial heuristics should be insufficient, and thus we expect that only models with robust length generalization can perform reliably. We find that C-RASP[pos] perfectly tracks length generalization results in our fine-tuning experiments: Every task which belongs to C-RASP[pos], and is thus predicted to length-generalize, does so, whereas tasks not in C-RASP[pos] do not length-generalize. Additionally, the Directional Bias of pretrained models disappears with fine-tuning, confirming the fact that it was an artifact of pretraining. We next seek to understand what leads to the development of this asymmetry by analyzing induction and anti-induction circuits (defined in Section 4.4) both in pretrained models, where this asymmetry exists, and in our fine-tuned models, where it doesn't.

## 4.4 Source of the Directional Bias: A Mechanistic Perspective

We hypothesize that the source of directional bias in pretrained models lies in the difference between the presence and strength of induction and anti-induction circuits as described in Section 2. We posit that, in pretrained models, anti-induction heads are less common, but that fine-tuning can boost these, removing the asymmetry. We use Unique copying (UF and UB) setups to evaluate our hypothesis.

**Induction and anti-induction heads are causal in both fine-tuned and pretrained models.** We conduct a patching experiment to confirm that removal of induction heads hurts performance in the UF task, while removal of anti-induction heads hurts performance in the UB task. Indeed, both in fine-tuned and pretrained models, removing induction heads is catastrophic for UF and has a negligible effect in UB, and removing anti-induction heads cripples backward copying while leaving forward copying intact. On fine-tuned checkpoints, the effect is amplified to the extreme: removing relevant heads drops the accuracy to zero, while removing another type of heads has no effect (Figure 7b). Thus, induction and anti-induction heads have a causal effect on model performance (Details of patching experiment in Appendix E). Additionally, the induction and anti-induction heads get amplified via fine-tuning (Figure 7a). Fine-tuning strengthens the heads that matter: induction heads gain attention weight for forward copy, anti-induction heads for backward copy.

---

[7] https://huggingface.co/karpathy/gpt2_1558M_final4_hf

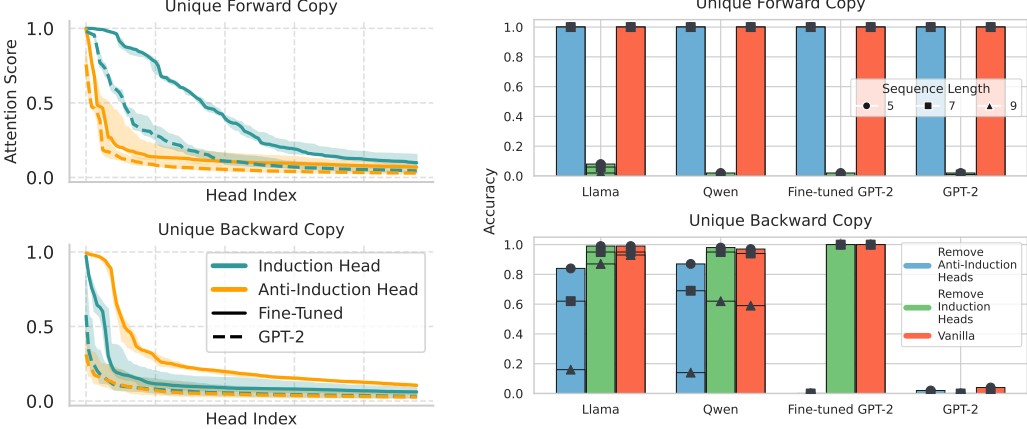

(a) Fine-tuning strengthens the relevant heads   (b) Induction and Anti-Induction heads are causal

Figure 7: (a) Top-10-percent attention heads ranked by their *attention score*, i.e. the mean attention weight that target-string tokens allocate to either the **same** source token (anti-induction) or the **next** source token (induction). Dashed lines: pretrained checkpoints; solid lines: fine-tuned. Shaded bands cover source lengths 3–10. Fine-tuning eliminates the pretraining imbalance; for forward copying (UF) fine-tuning strengthens induction heads, for backwards copying (UB) fine-tuning boosts anti-induction heads. (b) Patching experiment that ablates each family of heads while measuring task accuracy. For UF, removing induction heads (red → green) is fatal, whereas ablating anti-induction heads is inconsequential (red → blue); the roles reverse for UB. This holds across models & sequence lengths confirming the *causal* status of the two circuits.

> **Takeaway:** The directional bias in pretrained transformers can be explained by the difference in strength and prevalence of induction vs. anti-induction heads. Fine-tuning can be used to amplify either of these heads and thus remove the asymmetry.

## 5 Discussion

**Related Work.** *Length generalization* has been studied extensively [Anil et al., 2022, Abbe et al., 2023, Zhou et al., 2024], including attempts to improve it using scratchpads [Abbe et al., 2023, Hou et al., 2024] or specific positional embedding schemes [Press et al., 2022, He et al., 2024, Cho et al., 2024]. *Transformer limitations:* besides copying and retrieval, transformers struggle with problems including sensitive functions [Hahn and Rofin, 2024] and state tracking [Merrill and Sabharwal, 2023]. These limitations can be related to failures in LLMs on tasks such as multiplication [Frieder et al., 2024, Satpute et al., 2024, Amiri et al., 2025] and maintaining state information [Toshniwal et al., 2022, Kim and Schuster, 2023, Merrill et al., 2024, Zhang et al., 2025]. *Induction heads* have been studied extensively since Elhage et al. [2021] and their existence has been linked to in-context learning and copying [Olsson et al., 2022, Chen et al., 2024a, Singh et al., 2024, Crosbie and Shutova, 2025]. Retrieval heads [Wu et al., 2025] generalize this notion by conditionally copying information from arbitrary positions, subsuming our definitions of induction and anti-induction heads.

**Takeaways.** While failures in length generalization in copying and retrieval tasks may not affect typical user interaction, they pose reliability concerns in specialized applications. For instance, AI coding assistants like GitHub Copilot [Chen et al., 2021] routinely handle non-semantic identifiers (commit hashes, variable names, function names). Glitches in copying such strings could lead to bugs in the deployment of critical code and would therefore be highly undesirable. Pretraining of LLMs has been shown to impart biases, be it for specific tokens or specific entities [He et al., 2025] and in-context performance can be brittle based on the choice of tokens used. This can lead to reliability risks in domains requiring high precision such as medicine [Li and Chong, 2024, Bélisle-Pipon, 2024]. In contrast, the left-right asymmetry we uncover is indicative of a latent ability that is difficult to elicit in pretrained models without explicit fine-tuning. We ask practitioners to be aware of the

existence of such biases and consider task-specific fine-tuning to eliminate this kind of asymmetry if it is considered undesirable for a downstream application.

On the other hand, glitches in copying large semantically neutral text (SHA keys, DNA sequences) would fall under the umbrella of repeated copying. To achieve complete reliability and compensate for persistent architectural limitations, practitioners might need to resort to external tool usage in LLMs. Similar to how tool usage is used to compensate for mathematical weaknesses in LLMs [Gao et al., 2023, Chen et al., 2024b, Gou et al., 2024, Wang et al., 2024], designing principled policies that decide when to use tools like programmatic copy-pasting could be an interesting research direction.

**Limitations.** We focus on two open LLM model families, not covering the entire model landscape, especially closed source systems or models based on recurrent architectures, which might show different sets of abilities or limitations [Sarrof et al., 2024, Grazzi et al., 2025, Siems et al., 2025]. Second, due to resource limitations, fine-tuning experiments in Section 4.3 are limited to a single, 1.5B sized model. We also do not analyze the underlying pretraining data of the models to get an estimate of what the correct in-distribution bin should be for any of our models. Larger contexts should in principle push the out-of-distribution boundary to higher lengths as the in-distribution size increases. It remains an open question whether length generalization would even matter for larger models with a huge in-distribution context size. Finally, our theoretical results linking C-Rasp[pos] to length-generalizability in Huang et al. [2025] have only been proven formally for APE, and applicability to RoPE has only been tested empirically.

# 6 Conclusion

We asked whether large-scale pretraining can erase the transformer's native length-generalization limits, using a foundational family of retrieval and copying tasks as testbed. We find that pretraining boosts certain abilities, but transformers remain bound by the same length-generalization limits they were "born" with: Pretraining amplifies induction circuits, making right-/forward-oriented retrieval and copying notably easier, but does not overcome the difficulty inherent to non-unique retrieval and copying. Fine-tuning can rebalance induction and anti-induction heads and restore full, theory-aligned generalization, but only by explicitly teaching the model its missing capacities. Our results thus suggest that, while large-scale pretraining brings remarkable capabilities, it cannot fundamentally rewrite the architecture's core inductive biases.

## Acknowledgements

We thank anonymous reviewers for their encouraging and constructive feedback. This research is funded in part by the Deutsche Forschungsgemeinschaft (DFG, German Research Foundation) – Project-ID 232722074 – SFB 1102 "Information Density and Linguistic Encoding"; Project-ID 471607914 – GRK 2853/1 "Neuroexplicit Models of Language, Vision, and Action"; and Project-ID 389792660 – TRR 248 "Foundations of Perspicuous Software Systems". We would like to thank Mark Rofin, Blerta Veseli, Brian DuSell, Ekaterina Shutova, Kayo Yin, Xinting Huang, Yifan Wang for discussions and feedback on the draft.

## Contributions

MJ contributed to Sections 4.1, 4.2, 4.3, scaling experiments, paper refinement and the Appendix. YV contributed to Sections 4.1, 4.2, 4.3, 4.4, and paper refining. YS contributed to Sections 4.1, 4.2, the Appendix, scaling experiments, paper drafting and refining. SB contributed to Section 4.4. VD and EP provided feedback, supervision, and refined the paper. MH supervised the project and contributed to Section 3, paper drafting, and refining.

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

# A    Detailed Theoretical Proofs

## A.1    C-RASP[pos] as a Framework for Predicting Length Generalization

C-RASP[pos] (pos for "positional information") was introduced as C-RASP[periodic,local] by Huang et al. [2025][8] as a formalization of RASP-L and the RASP-L conjecture of Zhou et al. [2024]. It extends C-RASP [Yang and Chiang, 2024] by incorporating positional information. Huang et al. [2025] connected expressibility in C-RASP[pos] to length generalization for transformers both theoretically and empirically:

1. They proved that transformers, when trained with an idealized model of learning, generalize on problems expressible in C-RASP[pos].

2. Across a battery of algorithmic and formal language problems, they showed that problems expressible in C-RASP[pos] show length generalization when transformers are trained with SGD.

3. For problems in this battery that are not expressible in C-RASP[pos], they showed that length generalization was empirically unsuccessful.

These results unify and explain a variety of prior results; for instance, the link between C-RASP[pos] and length generalization explains why transformers show persistent glitches in FlipFlop (empirically observed by Liu et al. [2024a]), it explains why length generalization on copying is difficult in the presence of repetition [Zhou et al., 2024, Jelassi et al., 2023], and why length generalization on addition is difficult [Zhou et al., 2024]. Importantly, C-RASP[pos] offers a single framework for understanding length generalization, with well-understood methods for understanding expressivity. It is more fine-grained than various other methods for understanding transformers' expressivity: For instance, all tasks considered in Zhou et al. [2024], Huang et al. [2025] are in $TC^0$ and in principle expressible by transformers, but length generalization varies substantially in line with C-RASP[pos] expressivity. Thus, C-RASP[pos] provides a more fine-grained perspective by focusing on length generalizability.

C-RASP[pos] can thus be viewed as a formal model of the space of problems that transformers can represent across different input lengths and are likely to length-generalize on. transformers can in principle represent problems outside of C-RASP[pos], but these are then likely to not show length generalization because representations at increasing lengths will require increases in model size or parameter norm – if learning tends to find simpler solutions, one will expect length generalization to fail [Zhou et al., 2024, Huang et al., 2025]. For instance, one can construct a transformer performing copying over strings of any fixed length $N$, e.g. by hard-coding matching pairs of positions [Bhattamishra et al., 2024], but any construction that works across all lengths up to $\leq N$ will require a growth of model size (in a sense made formal by Huang et al. [2025]) as $N$ increases, i.e., length generalization is unlikely.

As Huang et al. [2025] formalizes the RASP-L conjecture of Zhou et al. [2024], we expect that the same conclusions would likely be reached using the RASP-L language from Zhou et al. [2024]; the advantage of C-RASP[pos] as compared to RASP-L is that its expressiveness is provably understood (including impossibility proofs), and is theoretically linked to length generalization (Theorem 7 in Huang et al. [2025]).

For self-containedness, we recapitulate the operations used in C-RASP[pos] here:

**Definition 3.** *Let $\Sigma$ be an alphabet.*

*Let $\Phi$ be a set of* unary relations $\phi : \mathbb{N} \to \{\bot, \top\}$ *where each $\phi$ satisfies*

- *(Periodicity) for some $\Delta > 0$, it holds that $\phi(i) = \phi(i + \Delta)$ for all $i$*

*Let $\Psi$ be a set of* binary relations $\psi : \mathbb{N} \times \mathbb{N} \to \{\bot, \top\}$ *here each $\psi(i, j)$ satisfies[9]*

- *(Depends only on Distance) $\psi(i, j)$ only depends on $i - j$*

- *(Restricted to Bounded Distance) $\{i : \psi(i, j) = \top\}$ is finite, for any $j \in \mathbb{N}$*

*Here, $\bot$ stands for "false" and $\top$ stands for "true".*

*A C-RASP[pos] program $P$ is defined as a sequence $P_1, \ldots, P_k$ of operations, each of the following types:*

---

[8]We use a shorter name just for convenience.

[9]These properties were referred to as *translation invariance* and *locality* in Huang et al. [2025].

|  | **Boolean-Valued Operations** |  | **Count-Valued Operations** |
|---|---|---|---|
| ***Initial*** | $P(i) := Q_\sigma(i)$ 
 for $\sigma \in \Sigma$ | ***Counting*** | $C(i) := \#[j \leq i, \psi(i,j)] \ P(j)$ 
 for $\psi \in \Psi \cup \{\top\}$ |
| ***Boolean*** | $P(i) := \neg P_1(i)$ 
 $P(i) := P_1(i) \wedge P_2(i)$ | ***Conditional*** | $C(i) := P(i) \ ? \ C_1(i) : C_2(i)$ |
| ***Constant*** | $P(i) := \top$ | ***Addition*** | $C(i) := C_1(i) + C_2(i)$ |
| ***Positional*** | $P(i) := \phi(i)$ 
 for $\phi \in \Phi$ | ***Subtraction*** | $C(i) := C_1(i) - C_2(i)$ |
|  |  | ***Constant*** | $C(i) := 1$ |
| ***Comparison*** | $P(i) := C_1(i) \leq C_2(i)$ |  |  |

The most important construct is the Counting operation, which returns an integer indicating for how many positions $j \leq i$ both $P(j)$ and $\psi(i,j)$ hold. In the special case where $\psi = \top$ (i.e., a predicate that always returns "true"), the operation just counts how often $P(j)$ holds for $j \leq i$. Another important operation is the Initial operation, where $Q_\sigma(i)$ is true if and only if the $i$-th symbol in the string equals $\sigma$. We refer to Huang et al. [2025] and Yang and Chiang [2024] for detailed definitions of the semantics.

The most important example for a function $\psi \in \Psi$ is the "predecessor" predicate checking if $j = i - 1$:

$$\psi(i,j) = \begin{cases} \top & \text{if } j = i - 1 \\ \bot & \text{else} \end{cases} \tag{1}$$

We will follow Huang et al. [2025], Yang and Chiang [2024] in using, as syntactic sugar, $\exists[j \leq i, \psi(i,j)]P(j)$ as a shorthand for $1 \leq \#[j \leq i, \psi(i,j)]P(j)$, and $\#[j \leq i] \ P(j)$ for $\#[j \leq i, \top] \ P(j)$.

In this paper, we are interested in tasks that, given a prefix, produce a token: either the overall output symbol in a retrieval task, or the next output symbol when copying a string. To formalize this, we assume that every program contains Boolean-valued operations named $\text{NEXT}_a(i)$ for each $a \in \Sigma$ such that $\text{NEXT}_a(i)$ holds if and only if the correct next symbol is $a$. Thus, we say that a task is expressible in C-RASP[pos] if there is a program defining $\text{NEXT}_a(i)$ for each $a \in \Sigma$ such that, for each $i$ in the span of the output (a single position for a retrieval task, the full output string for a copying task), $\text{NEXT}_a(i)$ holds (i.e, returns "true") at the final position $i$ if and only if the correct next symbol is $a$. Generation stops when none of these operations return "true".

## A.2 Expressiveness Results

We show:

**Lemma 4.** *UL, UR are expressible in* C-RASP[pos]*.*

*Proof.* A UR program is given in Huang et al. [2025]; the UL program is analogous; overall:

---
**C-RASP[pos] program for UR, UL**

$$\text{PRED}_a(i) := \exists[j \leq i, j = i - 1]Q_a(j) \tag{1}$$
$$\text{for each } a \in \Sigma$$
$$CBIGRAM_{ab} := \exists[j \leq i]Q_b(j) \wedge \text{PRED}_a(j) \tag{2}$$
$$\text{for each } a, b \in \Sigma$$
$$\text{NEXT}_a(i) := \bigvee_{\sigma \in \Sigma} [Q_\sigma(i) \wedge \text{CBIGRAM}_{\sigma a}(i)] \qquad \textit{(UR version)} \tag{3}$$
$$\text{for each } a \in \Sigma$$
$$\text{NEXT}_a(i) := \bigvee_{\sigma \in \Sigma} [Q_\sigma(i) \wedge \text{CBIGRAM}_{a \sigma}(i)] \qquad \textit{(UL version)} \tag{4}$$
$$\text{for each } a \in \Sigma$$

---

This program is a direct formalization of the (anti-)induction head circuit, and basic to the other C-RASP[pos] programs constructed below. The first line checks for each $a \in \Sigma$, at each position $i$, whether position $i - 1$ holds the symbol $a$. The second line checks, for each bigram $ab \in \Sigma \times \Sigma$ whether it appears at some position in the context up to the $i$-th position. The third line then states that, in UR, the correct next token is $a$ if and only if the bigram $\sigma a$ has appeared in the context, where $\sigma$ is the symbol at position $i$. The fourth line is the analogous instruction for UL: Here, $a$ is predicted if and only if the bigram $a\sigma$ appears in the context. $\qquad \square$

**Lemma 5.** *NRFirst and NLFirst are expressible in* C-RASP[pos].

*Proof.* We first show the construction for NRFirst; it builds on the construction from Lemma 4:

---

**C-RASP[pos] program for NRFirst**

$$\text{ISLEFTMOST}(i) := \bigvee_{a \in \Sigma} [Q_a(i) \wedge (\#[j \leq i]Q_a(j)) \leq 1] \tag{1}$$

$$\text{PRED}_a(i) := \exists[j \leq i, j = i - 1]\,[Q_a(j) \wedge \text{ISLEFTMOST}(j)] \tag{2}$$
$$\text{for each } a \in \Sigma$$

$$\text{CBIGRAM}_{ab} := \exists[j \leq i]\,[Q_b(j) \wedge \text{PRED}_a(j)] \tag{3}$$
$$\text{for each } a, b \in \Sigma$$

$$\text{NEXT}_a(i) := \bigvee_{\sigma \in \Sigma} [Q_\sigma(i) \wedge \text{CBIGRAM}_{\sigma a}(i)] \tag{4}$$
$$\text{for each } a \in \Sigma$$

---

where $\text{NEXT}_a(i)$ holds at the final position if and only if the desired completion is the symbol $a$. The first line in the program checks, at each position, if it is the leftmost representative of the symbol that it holds: That is, whether there is a symbol $a \in \Sigma$ that occurs at position $i$ but at no earlier position. For each $a \in \Sigma$, the second line checks, at position $i$, whether the preceding position $i - 1$ holds the symbol $a$ – simulating the operation of a Previous-Token head. Additionally it also uses the computations made in the first line to check whether position $i - 1$ was the leftmost representative with such a property. For each possible bigram $ab \in \Sigma \times \Sigma$, the third line checks whether it has appeared or not. Finally, the fourth line says that $a$ is the correct next token if and only if the current position holds a symbol $\sigma$ such that $\sigma a$ appears as a bigram in the context. This last line is the one that simulates the operation of the Induction head itself.

The program for NLFirst is very similar, it only differs in the fourth line:

---

**C-RASP[pos] program for NLFirst**

$$\text{ISLEFTMOST}(i) := \bigvee_{a \in \Sigma} [Q_a(i) \wedge \neg \exists[j \leq i]Q_a(j)] \tag{1}$$

$$\text{PRED}_a(i) := \exists[j \leq i, j = i - 1]\,[Q_a(j) \wedge \text{ISLEFTMOST}_a(j)] \tag{2}$$
$$\text{for each } a \in \Sigma$$

$$\text{CBIGRAM}_{ab} := \exists[j \leq i]\,[Q_b(j) \wedge \text{PRED}_a(j)] \tag{3}$$
$$\text{for each } a, b \in \Sigma$$

$$\text{NEXT}_a(i) := \bigvee_{\sigma \in \Sigma} [Q_a(i) \wedge \text{CBIGRAM}_{a\sigma}(i)] \tag{4}$$
$$\text{for each } a \in \Sigma$$

---

where $\text{NEXT}_a(i)$ holds at the final position if and only if the desired completion is the symbol $a$.

$\square$

**Lemma 6.** *NRLast and NLLast are not expressible in* C-RASP[pos].

*Proof.* We show this by reducing them to Lemma 36 in Huang et al. [2025]. Assume, for the sake of contradiction, that NRLast was expressible in C-RASP[pos]. This program would discriminate between the two languages

$$(w0|w1|i)^* w1i^* \langle sep \rangle w \qquad \text{(retrieving a ``1'')} \tag{5}$$

and

$$(w0|w1|i)^* w0i^* \langle sep \rangle w \qquad \text{(retrieving a ``0'')} \tag{6}$$

Such a program could be turned into a program discriminating between the two languages

$$(w0|w1|i)^* w1i^* \tag{7}$$

and

$$(w0|w1|i)^* w0i^* \tag{8}$$

By a simple transformation of the alphabet, this in turn would amount to discriminating between the two languages

$$(a|b|i)^* ai^* \tag{9}$$

and

$$(a|b|i)^* bi^* \tag{10}$$

As described in the proof of Lemma 36 in Huang et al. [2025], a C-RASP[pos] program discriminating between these two languages could be used to create a C-RASP[pos] program for recognizing the language

$$(a|b|i)^* bi^* b(a|b|i)^* \tag{11}$$

which in turn is impossible by Lemma 35 in Huang et al. [2025]. The same reasoning holds for NLLast. □

**Lemma 7.** *UF, UB are expressible in* C-RASP[pos].

*Proof.* This task can be viewed as iterated application of UL and UR, respectively, and can thus be done with the same program. The only complication is that (i) at the beginning of copying, at $\langle sep \rangle$, the program needs to retrieve the symbol that had followed $\langle bos \rangle$, and that (ii) generation should stop once $\langle sep \rangle$ has been generated a second time. These extra conditions can be encoded into C-RASP[pos]. □

**Lemma 8.** *NF, NB are not expressible in* C-RASP[pos].

*Proof.* Huang et al. [2025] (Theorem 12) show that NF is not in C-RASP[pos] by showing that all problems expressible C-RASP[pos] have logarithmic communication complexity in the model where Alice and Bob have access to the first and second halves of the string, respectively. Because NF has no sublinear communication protocol in this model, it is not in C-RASP[pos]. The same argument applies to NB. □

We conclude from all lemmas in this section that there is no theoretical difference between left and right versions of our retrieval and copying tasks:

**Corollary 9** (Repeated from Theorem 2). *Across all tasks, there is no* C-RASP[pos] *expressibility difference between R vs L versions, and F vs B versions.*

*Proof.* Immediate from the lemmas in this section. □

### A.2.1 Generalized reverse copying

Let the token alphabet be $\Sigma = \{\sigma_1, \ldots, \sigma_z\} \cup \{\square\}$ where $\square$ represents separators. $\Sigma_w = \Sigma \setminus \{\square\}$ are all tokens that can be part of a word. Define the language $L = (\Sigma_w^+ \square)^* \Sigma_w^+$. Every $x \in L$ can be written uniquely as $x = w_1 \square w_2 \square \cdots \square w_n$ with $w_i \in \Sigma_w^+$ (i.e. each $w_i$ represents a word.) and also as $x = t_1 \ldots t_m$, where each $t_i \in \Sigma$ and is the token level representation of the string. We can think of copying this string $x \in L$ in 2 ways.

$$\rho_{\text{tok}}(t_1 \ldots t_m) = t_m \ldots t_1, \qquad \rho_{\text{word}}(w_1 \square \cdots \square w_n) = w_n \square \cdots \square w_1.$$

$\rho_{\text{tok}}(t_1 \ldots t_m)$ is the same as our backward copying setup, while $\rho_{\text{word}}(w_1 \square \cdots \square w_n)$ more closely matches how a typical user would ever want to interact with an LLM to reverse a string if ever.

**Corollary 10.** $\rho_{\text{word}}$ *is not expressible in* C-RASP[pos].

*Proof sketch.* Theorem 5 states that Non-Unique Copy backwards which is the same as $\rho_{\text{tok}}$—lies outside C-RASP[pos] by a communication-complexity argument. To transfer this inexpressibility to $\rho_{\text{word}}$ consider the following reduction.

**Reduction.** Embed any token string $t_1 \ldots t_m \in \Sigma^*$ into $L$ by inserting separators:

$$t_1 \square t_2 \square \cdots \square t_m \ \in L.$$

Because each "word" now has length 1, applying $\rho_{\text{word}}$ yields

$$\rho_{\text{word}}(t_1 \square \cdots \square t_m) \ = \ t_m \square \cdots \square t_1,$$

and deleting the $\square$ symbols recovers $\rho_{\text{tok}}(t_1 \ldots t_m)$. Hence an algorithm for $\rho_{\text{word}}$ would give one for $\rho_{\text{tok}}$ with constant overhead, contradicting Theorem 5. □

**Intuition.** $\rho_{\text{word}}$ reverses a list of *chunks*, whereas $\rho_{\text{tok}}$ does the same when every chunk is a single token. When each word has length 1 the two coincide, so solving word-order reversal would solve Non-unique copy-backwards as a special case—which creates a contradiction, and thus $\rho_{\text{word}}$ should also not be solvable.

| Problem | Model Size | LR | Max Steps |
|---|---|---|---|
| UL, UR, UF, UB | 2 layer; 4 head; 64 dim | 1e-3 | 30k |
| NLFirst, NRFirst | 4 layer; 4 head; 64 dim | 1e-3 | 30k |
| NLLast, NRLast, NF, NB | 4 layer ; 4 head; 256 dim | 1e-4 | 30k |

Table 1: Hyperparameters for training with APE on transformers trained from scratch

| Problem | Model Size | LR | Max Steps |
|---|---|---|---|
| UL, UR, UF, UB | 2 layer; 4 head; 64 dim | 1e-3 | 30k |
| NLFirst, NRFirst | 2 layer; 4 head; 64 dim | 1e-3 | 30k |
| NLLast, NRLast, NF, NB | 4 layer ; 4 head; 256 dim | 1e-4 | 30k |

Table 2: Hyperparameters for training with RoPE on transformers trained from scratch

# B    From Scratch Training Experiments

Here, we validate theoretical predictions about the success of transformers in length generalization on all retrieval and copying tasks, by training small transformers *from scratch*. We particularly validate that results hold both for APE (as in GPT-2, as originally targeted by the theory of Huang et al. [2025]) and RoPE (as found in various other modern LLMs). We adopt the methodology from Huang et al. [2025] to demonstrate all our experiments when training transformers from scratch. Like Zhou et al. [2024], we sample independent *training batches* on the fly instead of using a finite-size training set. In contrast, each *test set* contains 2000 samples that are sampled at the beginning of each experiment.

We train models with the default CausalLMLoss from the transformers library, the length of inputs is sampled uniformly from minimum up to maximum length in the specified range. This is true for training data where the range is $(l_{min}, 50)$ where $l_{min}$ is the minimum length possible for a given task (varies from 2 - 4, depending on the task variant and the minimum length of characters required to make the string valid). Our test sets have the size – $(l_{min}, 50), (51, 100), (101, 150)$. At each step, the model outputs the correct continuation – for retrieval the correct token to be retrieved, for copying the subsequent next token from the input sequence. The models are trained on a whole sequence of tokens. We train decoder-only transformer from scratch, using implementations from HuggingFace transformers for APE (GPT2LMHeadModel) and RoPE (LlamaForCausalLM). We train models for maximum 30K steps with a batch size of 64. We stop training early once the model's accuracy reaches 100% on the in-distribution test set (the one in range $[l_{min}, 50]$. The model is trained with a dropout rate of 0.0, and we use AdamW, with a weight decay rate of 0.01 (choices based on Huang et al. [2025]).

For experiments with APE, at training time, we add random offsets to position indices so that all position embeddings are trained (following Huang et al. [2025]). The offsets are sampled uniformly at random in the range $[0, 150 - \ell_{curr}]$ where $\ell_{curr}$ is the length of the current string. For RoPE, such an offset is not required in training[10]; instead a scaling factor of 32.0 is set; the RoPE scaling type used was linear.

In preliminary experiments, we found that different model architectures, while achieving 100% accuracy on in-distribution data, may perform differently on out-of-distribution data. To draw a conclusion about how the model performs on a problem in general, we determine the hyperparameters as follows: We consider configurations of {1, 2, 4} layers, {1, 2, 4} heads and model dimension of {16, 64, 256}, and learning rate of {0.001, 0.0001}. We sweep all the configurations by iterating over every combination and choose the one that achieves the highest accuracy on $[51, 100]$ among those configurations whose accuracy on $[l_{min}, 50]$ is 100%. When there are multiple such options, e.g., their accuracy on $[51, 100]$ is 100%, the one with the simplest architecture is selected (when estimating complexity, we assume the following priority: number of layers > number of heads > model dimension). The final hyperparameters we used for each task are shown in Table 1 and 2. After we determine the hyperparameter configuration, we run the experiments with multiple random seeds and report the average accuracy of 3 successful runs (those runs where the model achieves 100% accuracy on in-distribution data).

**UF, UB vs UR, UL:**    While training UF and UB with RoPE, we did not observe perfect length generalization when directly training on the task, unlike what we observed with APE. However, we note that UF and UB can be thought of as repeated applications of UR and UL, for which we did obtain perfect length generalization even with RoPE – showing that length-generalizable RoPE transformers do exist for UF and UB.[11] We thus use the RoPE transformers for UR and UL to model UF and UB. We take separate versions of transformers on UR and

---

[10]In fact, relative positional encodings such as RoPE are by definition invariant to the addition of a constant offset to the indices

[11]We leave to future research to determine whether this phenomenon reflects some deeper discrepancy between RoPE and APE, or superficial optimization difficulties.

UL (without any separator before the query token) and autoregressively use the UR and UL models to generate a full continuation till an EOS is reached to get correct solutions for UF and UB. Therefore we format the input as: $X = \langle bos \rangle \langle sep \rangle x_1 \ldots x_N \langle eos \rangle \langle sep \rangle$. Upon seeing the separator, the model predicts $x_1$, and then $x_2$ and so on till we get to $x_N$, and then an $\langle eos \rangle$ is predicted. We do the same with UL to get predictions for UB.

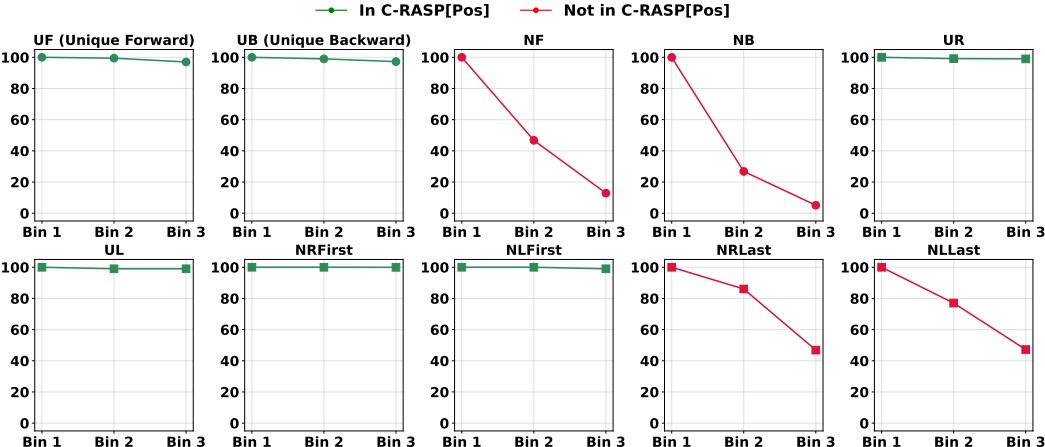

Figure 8: Transformers trained from scratch with APE, perfectly aligns with C-RASP[pos] predictions.

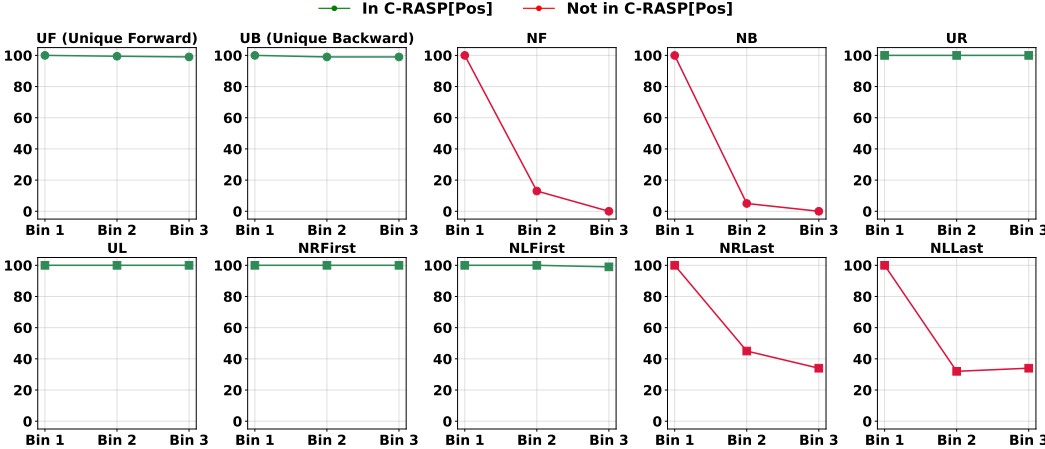

Figure 9: Transformers trained from scratch with RoPE, aligns with C-RASP[pos] predictions, even though the length generalizability guarantees given by C-RASP[pos] only apply to APE. It should be noted that the accuracy on UF, UB plotted here are not obtained by directly training, but by applying UR, UL repeatedly as described in the paragraph B. The bins here correspond to the evaluation bins of sizes $(l_{min}, 50), (51, 100), (101, 150))$.

## C    Additional experimental details and results

### C.1    Prompting: In-Context Retrieval Templates (Section 4.1)

Table 3 below lists the three prompt factors and their possible values. A Cartesian product of all prompt settings across the 3 categories given below was taken to generate 20 prompt templates.

**Separator Examples**

The input string is given in the following format based on the category.

- SEP : $x_1, x_2 \ldots x_{n-1} \| x_n$
- NOSEP : - $x_1, x_2 \ldots x_{n-1} x_n$

## Table 3: Prompt Template Grid Factors

| Factor | Options |
|--------|---------|
| Separator | SEP, NOSEP |
| Few-shot | SMALL, SAME |
| Template | BARE, SIMPLE_RULE, SIMPLE_RULE_EXPL., MATH_RULE MATH_RULE_EXPL. |

**Few-Shot Pools**

- SMALL — Examples drawn from a held-out pool with lengths smaller than the input string $X$.

- SAME — Examples sampled from the same evaluation set as the input string $X$ (excluding $X$ itself).

**Instruction Templates**

The different possible instruction templates for each of the tasks are given below. In every case, the prompt ends with the unresolved query line, prompting the model to emit the answer token.

**BARE**    This template provides only the few-shot examples without any additional instructional or explanatory text. It requires the completion model to infer the task directly from the presented examples. Example for $UR$:

**SIMPLE_RULE**    The task is introduced through a concise English-language description defining the resolution rule. The idea of the setup is the test whether the model can follow explicit instructions based on linguistic clarity. The format across the tasks is as follows.

```
Each line is written as 'context|query: target'.
The vertical bar '|' separates the context from the query token.
All strings below follow this rule: {rule_simple}
{examples}
```

## Table 4: Simple Rules for each of the tasks

| Task | Template |
|------|----------|
| UL | The answer is the token immediately to the left of the single instance of the query token |
| UR | The answer is the token immediately to the right of the single instance of the query token. |
| NLLast | When the query appears multiple times, the answer is the token just to the left of its last appearance. |
| NRLast | When the query appears multiple times, the answer is the token just to the right of its last appearance. |
| NLFirst | When the query appears multiple times, the answer is the token just to the left of its first appearance. |
| NRFirst | When the query appears multiple times, the answer is the token just to the right of its first appearance. |

**SIMPLE_RULE_EXPLAINED**    This template enhances the simple English rule format by providing an additional worked example explanation. It begins by stating the simple rule, then illustrating through an example how the rule is actually applied. Following the same, the model is presented with some few-shot examples to reinforce the rule understanding.

**MATH_RULE**    This template explicitly formulates the task using formal mathematical definitions and notation. It introduces variables to clearly define the input string, context, query token, token indices, and continuation token. The mathematical notation precisely indicates conditions under which the continuation token should be selected, leaving no ambiguity regarding the required solution path. The template is as follows.

```
Let $X = x_1 \ldots x_n$ with $n \ge 4$ and $x_i \in \Sigma$ (token vocabulary).
The final token $x_n$ is the query token $q$. In the context $x_1 \ldots x_{{n-1}}$,
$q$ appears $t$ times at indices $q_1, \ldots, q_t$ ($1 \le q_1 \le q_t \le n-1$).
Continuation token $x_{{n+1}}$ is defined by: {rule_math}
Examples:
{examples}
```

**MATH_RULE_EXPLAINED**    Similar to the SIMPLE_RULE_EXPLAINED, this template begins with a detailed mathematical formulation of the task and is then followed by a worked-out example and then finally some few-shot set of strings.

Table 5: Math Rule Template

| Task | Template |
|------|----------|
| UL | $t = 1$ and $x_{n+1} = x_{q_1-1}$ |
| UR | $t = 1$ and $x_{n+1} = x_{q_1+1}$ |
| NLLast | $t > 1$ and $x_{n+1} = x_{q_t-1}$ |
| NRLast | $t > 1$ and $x_{n+1} = x_{q_t+1}$ |
| NLFirst | $t > 1$ and $x_{n+1} = x_{q_1-1}$ |
| NRFirst | $t > 1$ and $x_{n+1} = x_{q_1+1}$ |

## C.2  Prompting: In-Context Copying Templates (Section 4.1)

**Template variants.**  For our In-context Copying task suite, we use three lightweight prompt styles. There is no Cartesian grid as in retrieval, as each of the prompts had statistically little variance, and eliciting this ability from our LLMs was not as challenging as in retrieval.

- `BARE` — few-shot examples only, no instructional text.
- `OBEY` — a one-line English rule (`rule_simple`) introducing the few-shot block.
- `HINT` — a short hint (`rule_hint`) that highlights the input–output relation; mainly useful for backward copying.

**Rule strings injected by each template.**  The phrases referenced above are drawn from Table 6. Note that unique and non-unique regimes share identical wording because the operation (copy vs. reverse) is independent of token repetition once the input is fixed.

Table 6: `rule_simple` phrases for the four copying tasks.

| Task | `rule_simple` |
|------|---------------|
| UF | The output is exactly the same sequence as the input. |
| UB | The output is the input sequence written in reverse order. |
| NF | The output is exactly the same sequence as the input. |
| NB | The output is the input sequence written in reverse order. |

The corresponding `rule_hint` strings swap the lead-in "The output is . . . " with "In every example the output . . . " to make the instruction less imperative.

**Few-shot sampler.**  All copy prompts use the same held-out $k=5$-shot pool (length $L=5$, $N=1500$) that is disjoint from the evaluation set. Each prompt is built by picking five random lines from this pool, followed by the query built from the current test string.

**End-to-end prompt bare template example (UF)**

```
<bos> r m a j s : r m a j s <eos>
<bos> H w s x n : H w s x n <eos>
<bos> Q o F G J : Q o F G J <eos>
<bos> O a Y M F : O a Y M F <eos>
<bos> m I N r D : m I N r D <eos>
<bos> Z i b E B :
```

The unresolved line after  is the model's query; generation proceeds with greedy decoding ($T = 0$).

## C.3  Longer Vocabulary (Word-Level) Experiments (Section 4.1)

To ensure that the Uniqueness and Directional Biases observed in our primary character-level experiments are not simply artifacts of a synthetic setup, we conducted a parallel set of experiments using a more naturalistic word-level vocabulary. In case of LLMs, the distinction between a token-level and a word-level task may be less sharp than it appears. Kaplan et al. [2025] shows that LLMs form an "inner lexicon" where simple or common words are processed as cohesive semantic units. Given their results, this word-level task formulation can be considered analogous to the character-level task.

**Experimental Setup**  The setup for the word-level vocabulary mirrors the main experiments in Section 4.1 but replaces the character-based vocabulary with a vocabulary of 300+ unique English words. This allowed us to construct longer, more complex sequences with lengths up to 300 tokens. All other aspects of the methodology, few-shot prompting strategy, and task structures (e.g. UR vs. UL, UF vs. UB) remain identical. The exact vocabulary we used to generate the test strings in all of the experiments is as follows.

**Character-level vocabulary**

```
a, à, â, ā, ä, ą, b, c, ç, ć, ĉ, ċ, č, d, e, é,
ê, ë, ě, ē, ę, f, g, ĝ, ğ, h, ĥ, i, í, j, ĵ, k,
l, m, n, o, ó, ô, ö, õ, p, q, r, s, t, u, ú, ù,
û, ü, v, w, x, y, z,
A, Á, Â, Ã, Ä, Å, Æ, B, C, Ç, Ć, Ĉ, Ċ, Č, D, E,
È, É, Ē, Ė, F, G, H, I, Í, Ĩ, Î, Ï, Ī, Ĭ, Į, J,
K, L, Ĺ, M, N, Ń, O, Ō, Ô, Ö, P, Q, R, S, T, U,
Ú, Ù, Û, Ü, V, W, X, Y, Z,
0, 1, 2, 3, 4, 5, 6, 7, 8, 9
```

**Word-level vocabulary**

```
apple, ant, arrow, anchor, artist, animal, angle, apricot, arch, armor, axis, avenue
ball, bat, book, bridge, bottle, bucket, bench, bread, bell, button, brush, branch
cat, car, cup, cloud, clock, candle, coin, chair, circle, crown, castle, cookie
dog, door, desk, drum, duck, doll, diamond, dish, dress, dream, drop, dust
egg, ear, eye, engine, elbow, earth, envelope, exit, echo, edge, event, energy
fish, fan, fork, flower, flag, feather, fire, frame, forest, farm, fruit, fence
goat, game, glass, glove, gate, garden, gift, grape, guitar, gold, gear, group
hat, hand, horse, house, hill, hammer, heart, honey, hook, horn, hug, hope
ice, iron, ink, island, idea, image, item, ivory, icon, input, issue, idol
jar, jam, jet, jewel, jungle, jacket, juice, job, joke, joy, judge, jump
kite, key, king, knee, kitchen, knife, kitten, knight, kick, kettle, kind, koala
lamp, leaf, lion, lock, ladder, lake, lemon, line, letter, lip, light, lunch
man, map, moon, milk, mouse, mirror, mountain, market, meal, music, magnet, match
net, nose, nest, name, nail, night, number, note, neck, nurse, noise, nation
owl, oil, oven, orange, ocean, order, orbit, open, option, owner, object, office
pen, pig, pot, plate, plane, pumpkin, pearl, park, path, piano, point, paper
queen, quill, quiz, quilt, quiet, quick, quote, quest, queue, quake, quart, quark
rat, ring, rain, river, rope, road, rose, rock, rule, room, root, radio
sun, sock, star, ship, shoe, stone, sugar, song, salt, sand, seed, snake
top, toy, tree, train, table, tube, tiger, tool, time, tent, team, towel
urn, use, unit, uncle, under, upper, uniform, union, urban, urge, ultra, usual
van, vase, veil, voice, valley, visit, value, vest, vote, view, vine, victory
wax, web, wall, wind, water, wheel, wave, wolf, wing, worm, word, wood
xylophone, xerox
yam, yard, yarn, yawn, year, yellow, yogurt, yolk, youth, yield, yeti, yoga
zoo, zebra, zone, zero, zip, zinc, zeal, zest, zigzag, zoom, zombie, zodiac
```

## C.4   Additional results and plots for in-context prompting (Section 4.1)

### C.4.1   Additional results: character-level vocabulary

As evident from all the figures here (smaller completion models - Figure 10), (smaller instruct models - Figure 11), (bigger instruct models - Figure 12), the Directional Bias persists across models sizes and types. While in the smaller models the gaps smaller for the retrieval tasks, nevertheless there is still a pronounced gap within the copying task variants.

Amongst the prompts there was high variability for all our retrieval setups, but close to none for the copying setups. Our bare template performed the worst in most retrieval setups, and the highest accuracy metrics were achieved when the instruction template was – MATH_RULE_EXPLAINED, with the few shot setting as SMALL with the presence of a separator. For copying, just the BARE setup was enough to get high accuracy numbers without the need to specify any instructions (especially for the forward copying cases). Providing instructions did improve the performance especially for the Qwen2.5-Instruct models.

We also carried out the same experiments with newer Qwen3 [Team, 2025] models with "thinking" variants to see if our findings still hold. The results from Figures 13, 14, 15, 16 show that even the model trained to explicitly generate a chain-of-thought for solving task still have the same Directional and Uniqueness biases.

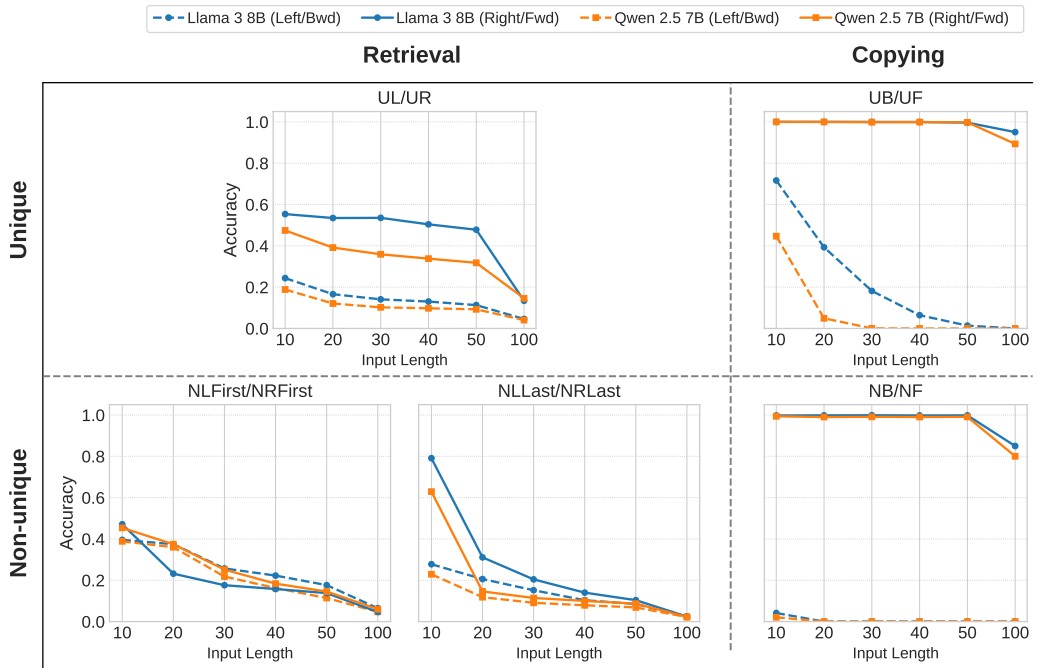

Figure 10: In-context accuracy for `Qwen2.5-7B` and `Llama-3-8B` across all our **character-level** tasks averaged over 3 seeds and prompt variations per task. The same patterns as discussed in Section 4.1 are visible.

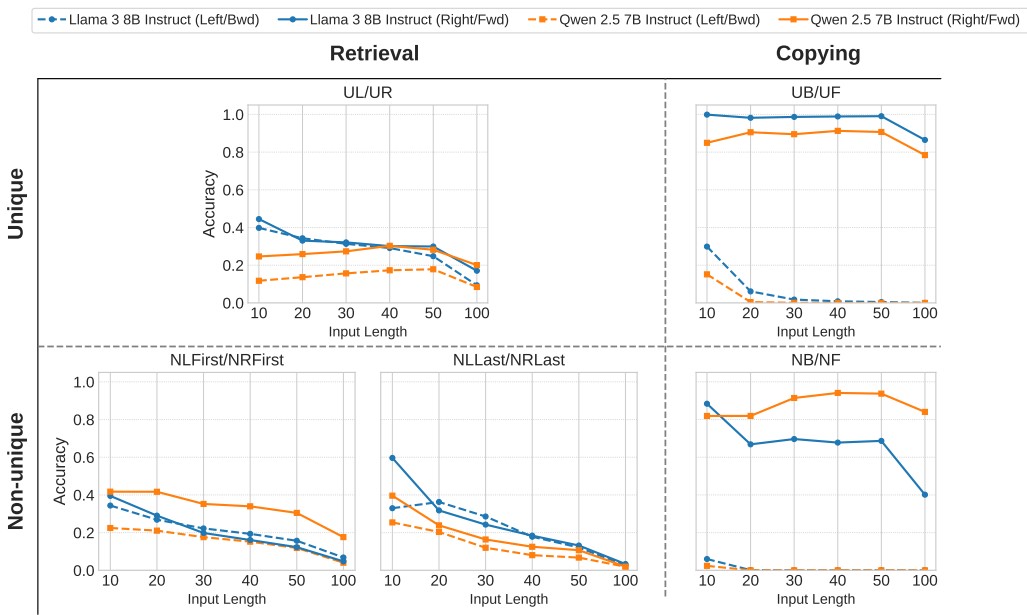

Figure 11: In context accuracy for `Qwen2.5-7B-Instruct` and `Llama-3-8B-Instruct` across all our **character-level** tasks averaged over 3 seeds and prompt variations per task. The same patterns as discussed in Section 4.1 are visible.

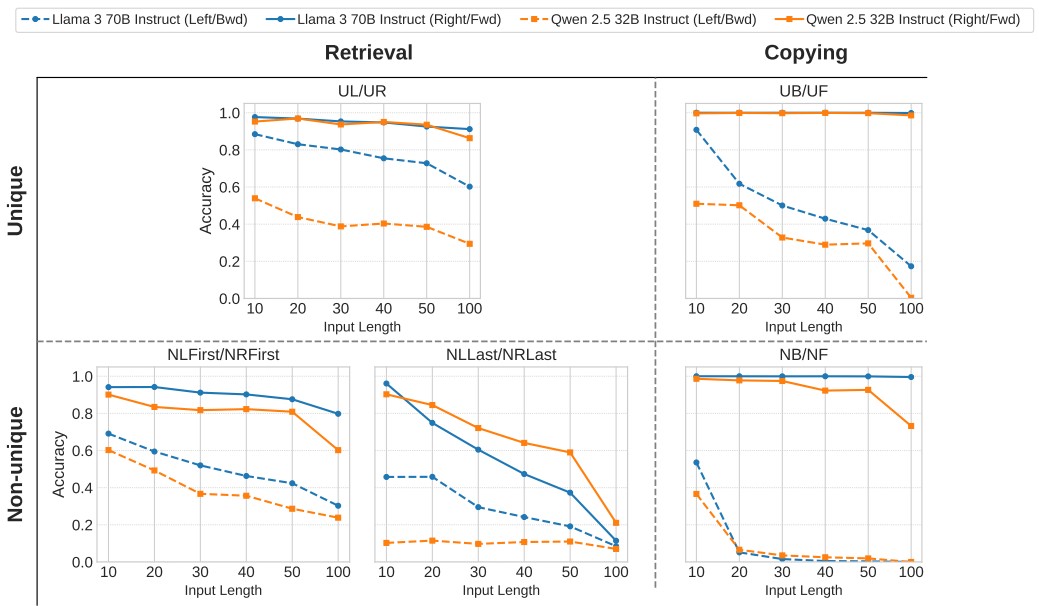

Figure 12: In context accuracy for `Qwen2.5-32B-Instruct` and `Llama-3-70B-Instruct` across all our **character-level** tasks averaged over 3 seeds and prompt variations per task. The same patterns as discussed in Section 4.1 are visible.

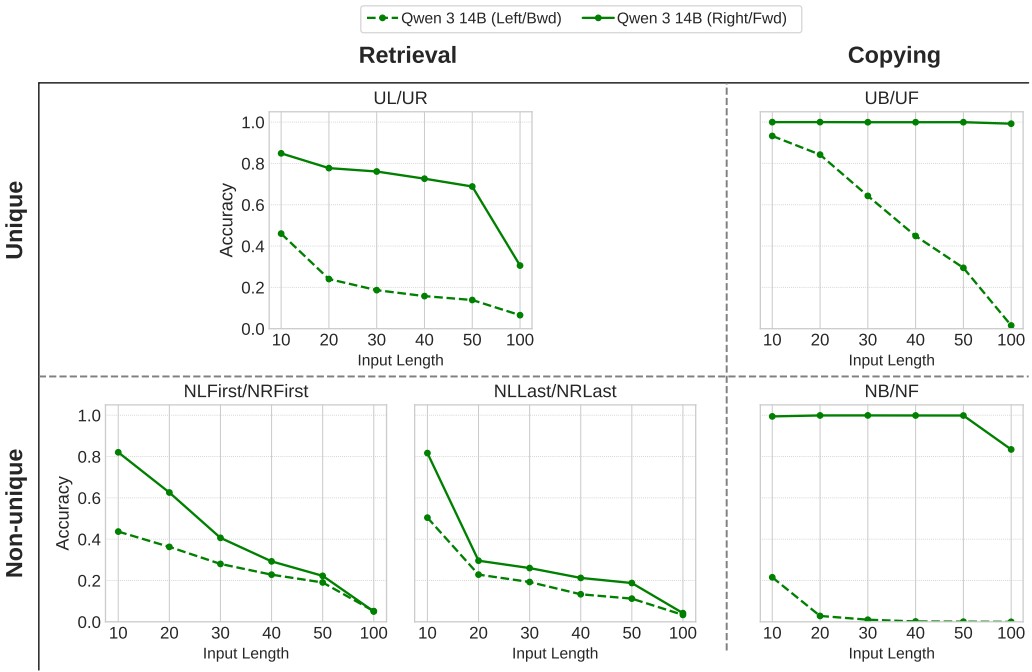

Figure 13: In-context accuracy for `Qwen3-14B` across all our **character-level** tasks averaged over 3 seeds and prompt variations per task. The same patterns as discussed in Section 4.1 are visible.

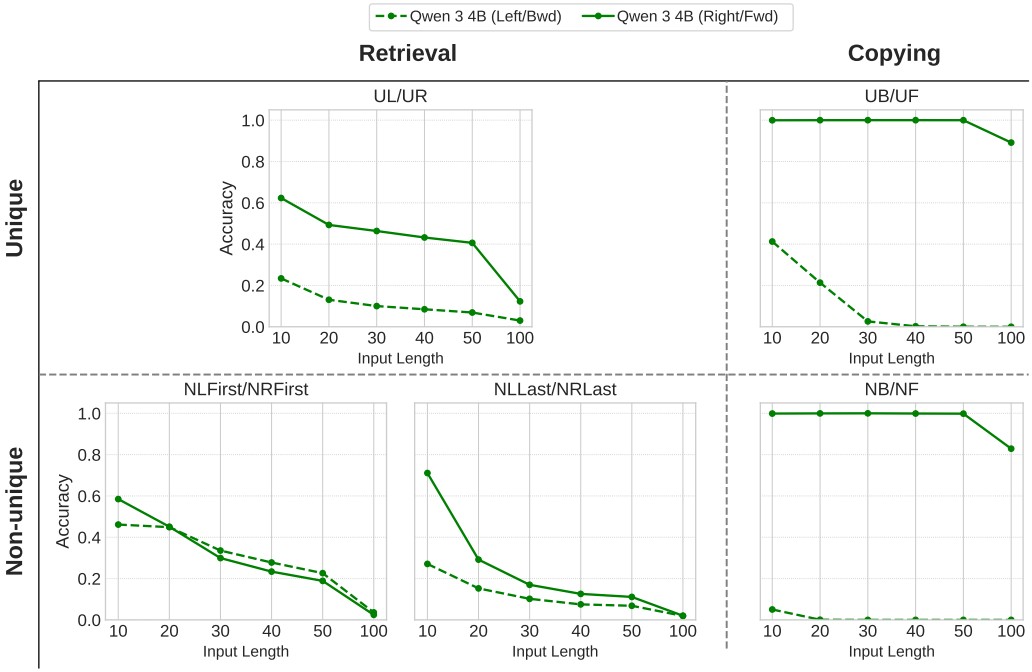

Figure 14: In-context accuracy for `Qwen3-4B` across all our **character-level** tasks averaged over 3 seeds and prompt variations per task. The same patterns as discussed in Section 4.1 are visible.

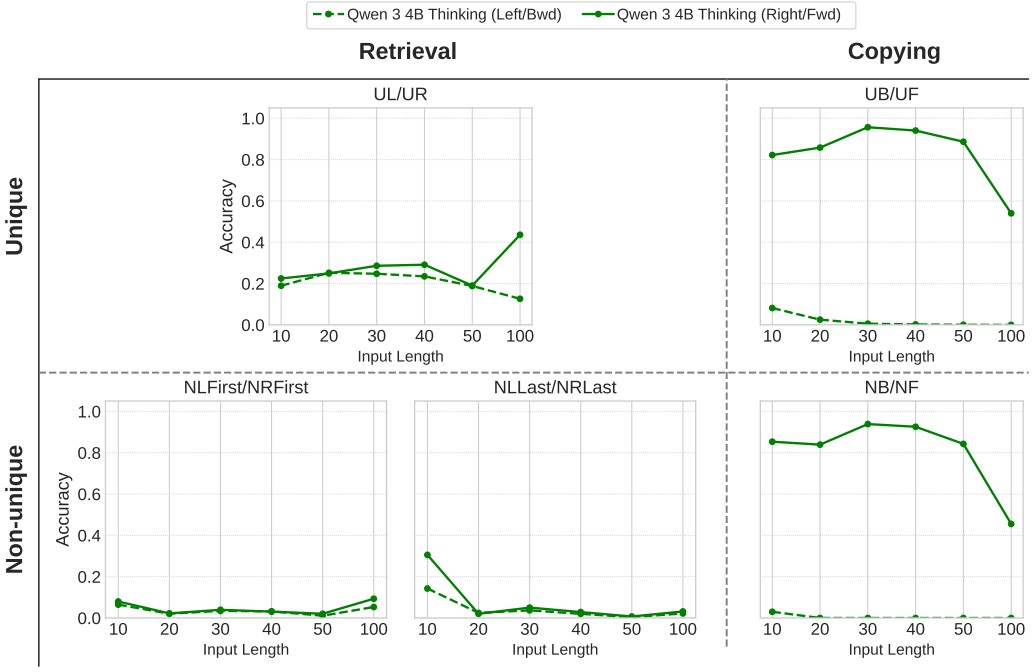

Figure 15: In context accuracy for `Qwen3-4B-Thinking` across all our **character-level** tasks averaged over 3 seeds and prompt variations per task. The same patterns as discussed in Section 4.1 are visible.

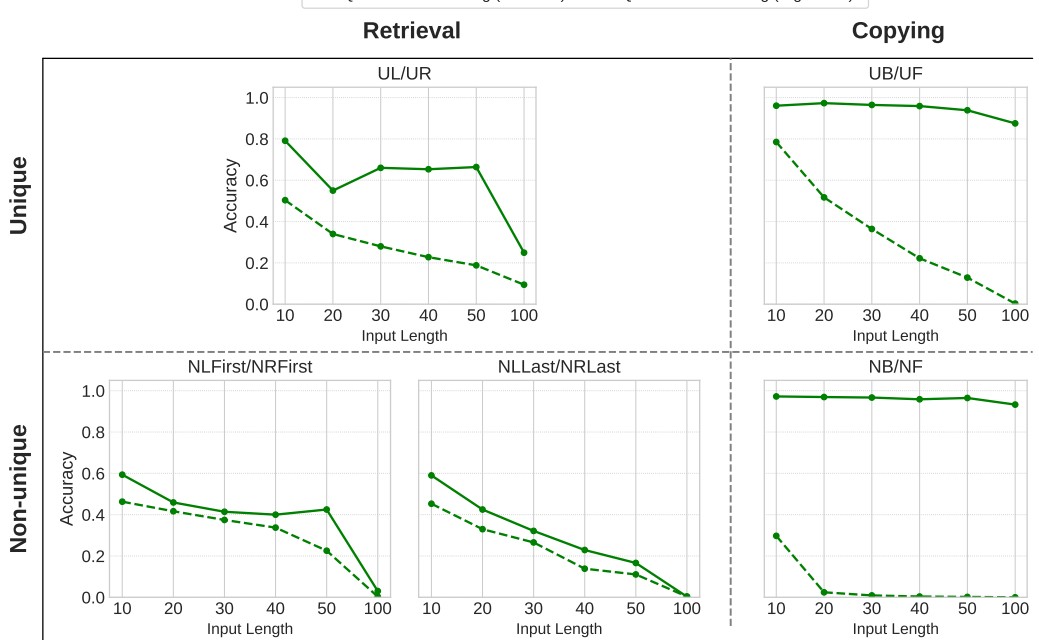

Figure 16: In context accuracy for `Qwen3-14B-Thinking` across all our **character-level** tasks averaged over 3 seeds and prompt variations per task. The same patterns as discussed in Section 4.1 are visible.

### C.4.2 Results: word-level vocabulary

We find that the core asymmetries persist even in the more naturalistic word-level setting (see Figures 17, 18, 19, 20, 21, 22, 23, 24). Although length generalization improves slightly for the larger models, the pretraining induced Directional Bias and the architecture-driven Uniqueness Bias remain clearly evident across all LLMs.

### C.5 Promping: Lorem Ipsum Copying (Section 4.2)

**Dataset generation.**    To stress-test copying while keeping text realistic, we construct a 1,500-example corpus of synthetic *Lorem Ipsum* paragraphs. Generation starts from one base paragraph produced by the `lorem` Python library and then applies light, probabilistic perturbations to increase the non-determinism of the texts.

- **Sentence count.** Each sample contains exactly 45 sentences (average ~350 tokens), enforced by iterative expansion/shuffling until the target length is reached.
- **Word/discourse noise.** With probability 0.3 a random sentence is repeated up to four times; with probability 0.5 random words inside a sentence are duplicated; with probability 1.0 words within a sentence are shuffled (except the leading capital).
- **Token budget.** Sequences are truncated to 500 BPE tokens to stay within context limits while preserving paragraph-level coherence.

**Prompt variants.**    We identified successful prompts for copying naturalistic text as follows. We downloaded research papers put up on ArXiv in April 2025, and checked if the prompt results in perfect accuracy (not even a single mistake). Our motivation was that ArXiv text includes challenging text with equations, and, due to recency, should not have appeared in the LLM training data. More specifically, we segregated the sections of the papers into paragraphs, making sure the total length of the paragraph to be copied has close to 500 tokens (while ensuring that we do not end the paragraph in the middle of a sentence). We then pre-validated our prompts on a dataset containing 500 samples each containing 500 tokens, and even the smaller models we test – `Llama3.1-8B` and `Qwen2.5-7B` perfectly copied the given text (not a single mistake). We proceeded with our 3 minimal prompts, shown in Table 9, which achieved perfect copying accuracy. Each of our prompts ends with an unresolved `<start>` token, prompting the model to emit a verbatim repetition of the preceding paragraph.

**Ratio of Ambiguous vs. Unambiguous Tokens.** Table 7 reports, for each model, the average number of ambiguous (tokens with multiple possible continuations) and unambiguous (tokens with a single continuation)

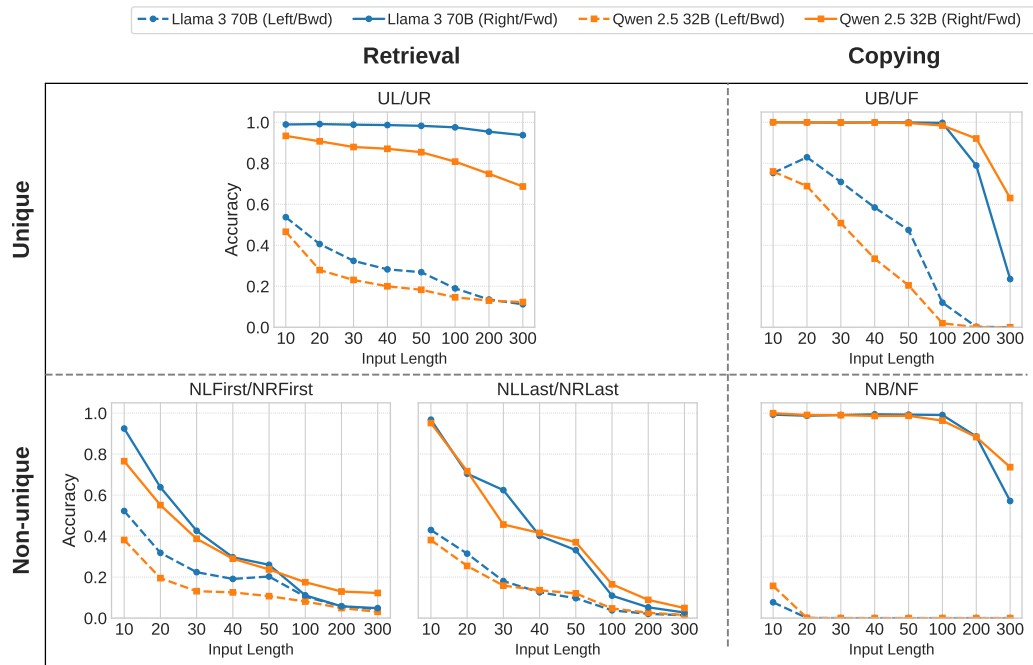

Figure 17: In context accuracy for `Qwen2.5-32B` and `Llama-3-70B` across all our **word-level** tasks averaged over 3 seeds and prompt variations per task. The same patterns as discussed in Section 4.1 are visible.

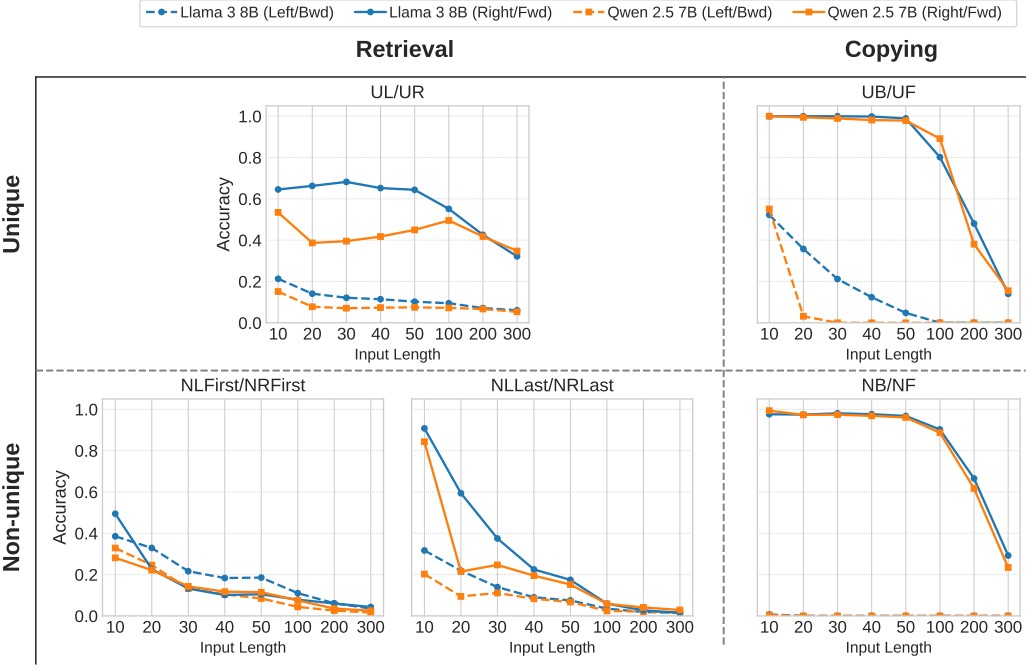

Figure 18: In-context accuracy for `Qwen2.5-7B` and `Llama-3-8B` across all our **word-level** tasks averaged over 3 seeds and prompt variations per task. The same patterns as discussed in Section 4.1 are visible.

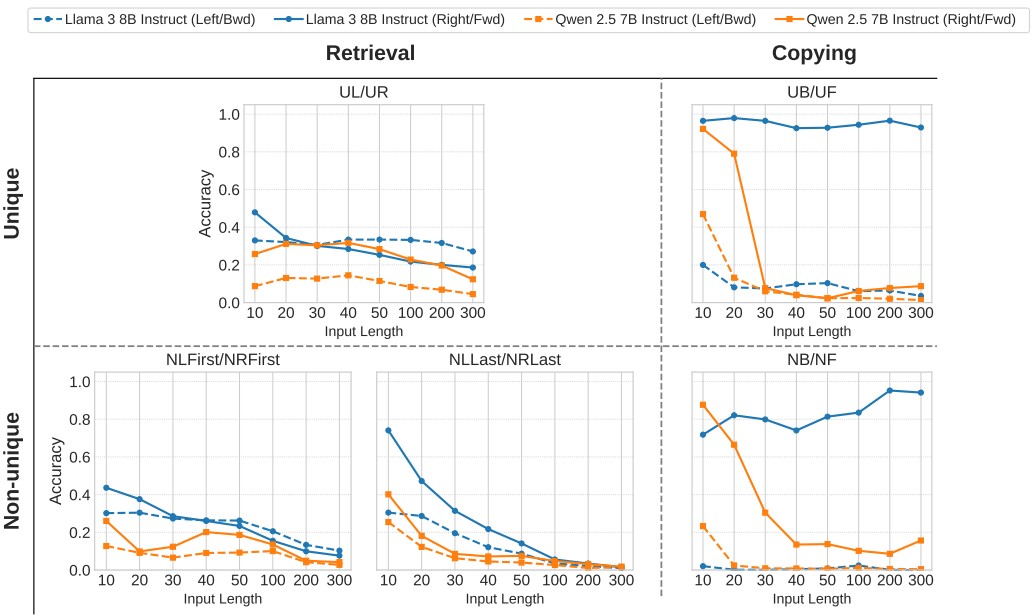

Figure 19: In context accuracy for `Qwen2.5-7B-Instruct` and `Llama-3-8B-Instruct` across all our **word-level** tasks averaged over 3 seeds and prompt variations per task. The same patterns as discussed in Section 4.1 are visible.

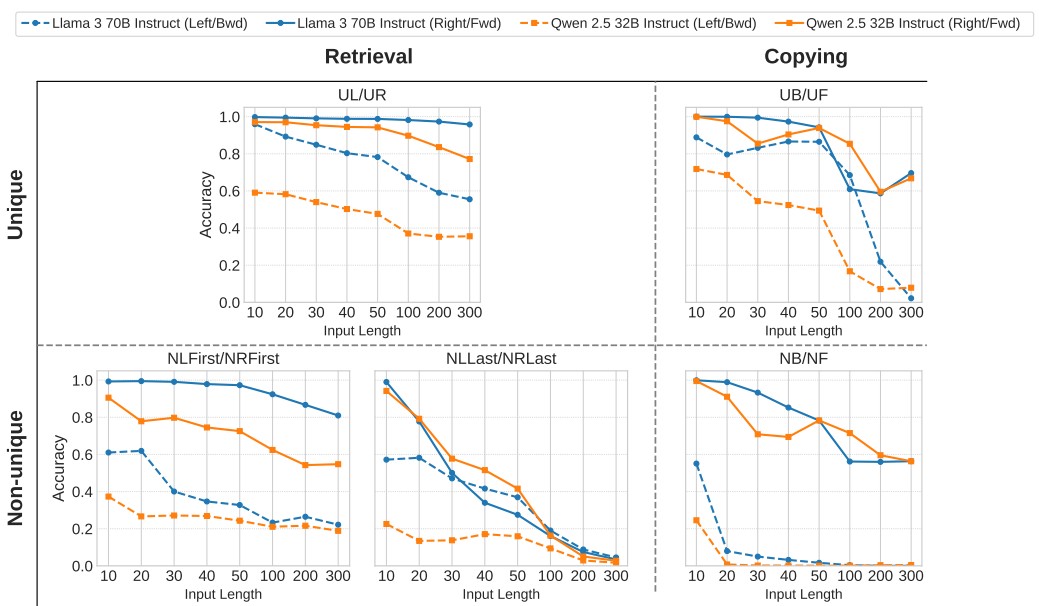

Figure 20: In context accuracy for `Qwen2.5-32B-Instruct` and `Llama-3-70B-Instruct` across all our **word-level** tasks averaged over 3 seeds and prompt variations per task. The same patterns as discussed in Section 4.1 are visible.

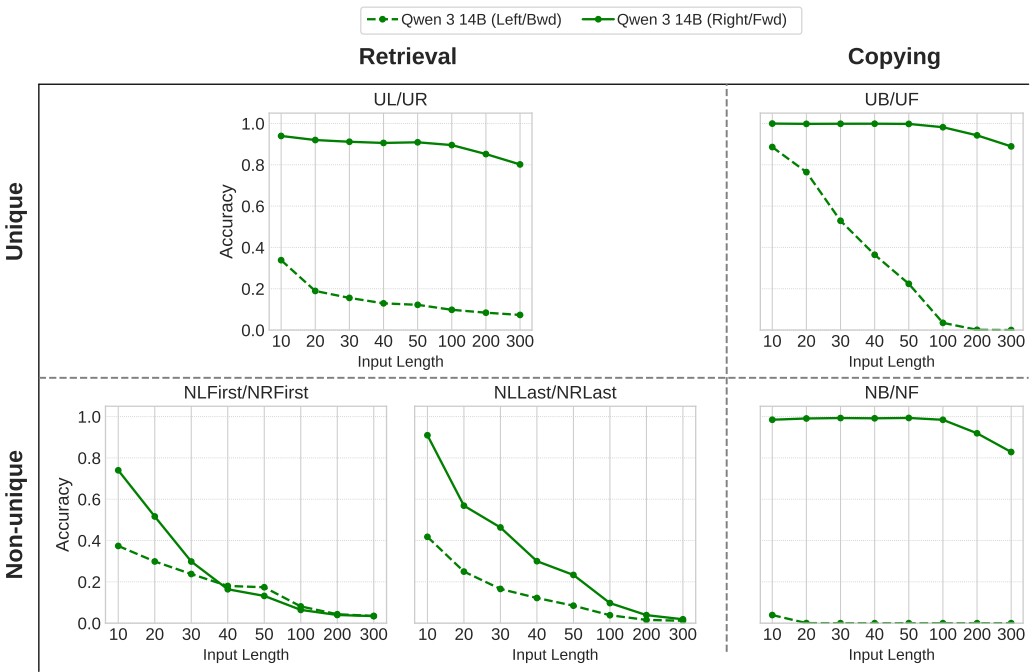

Figure 21: In-context accuracy for `Qwen3-14B` across all our **word-level** tasks averaged over 3 seeds and prompt variations per task. The same patterns as discussed in Section 4.1 are visible.

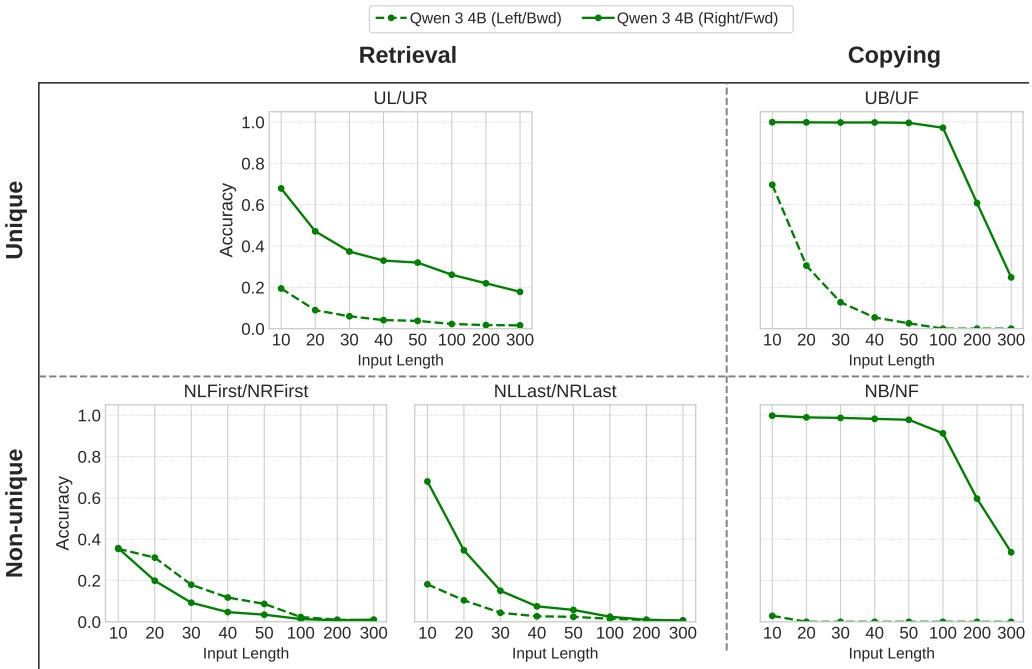

Figure 22: In-context accuracy for `Qwen3-4B` across all our **word-level** tasks averaged over 3 seeds and prompt variations per task. The same patterns as discussed in Section 4.1 are visible.

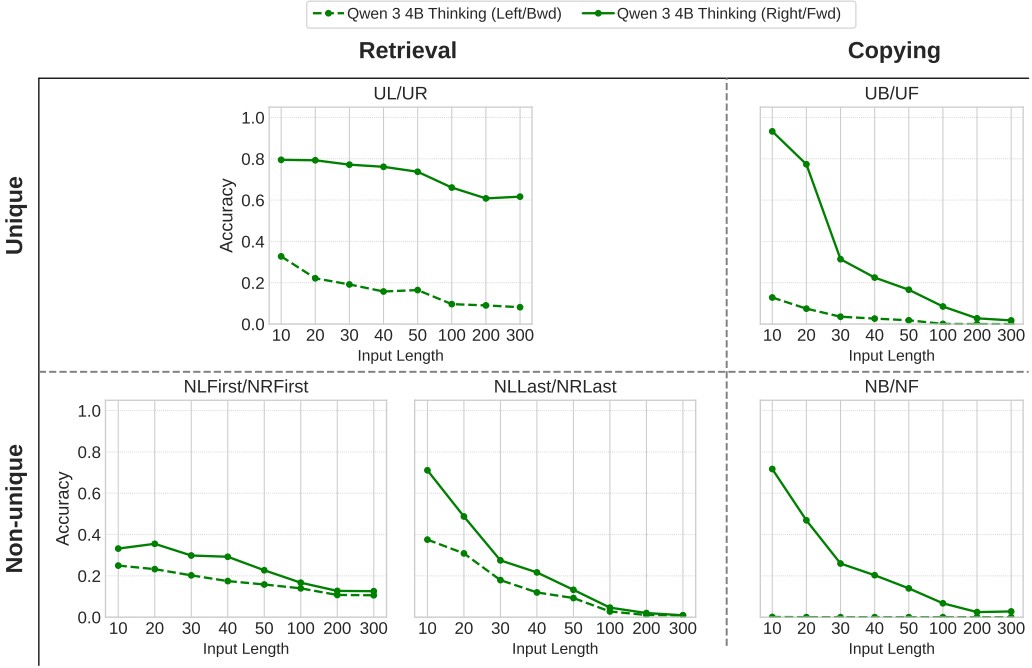

Figure 23: In context accuracy for `Qwen3-4B-Thinking` across all our **word-level** tasks averaged over 3 seeds and prompt variations per task. The same patterns as discussed in Section 4.1 are visible.

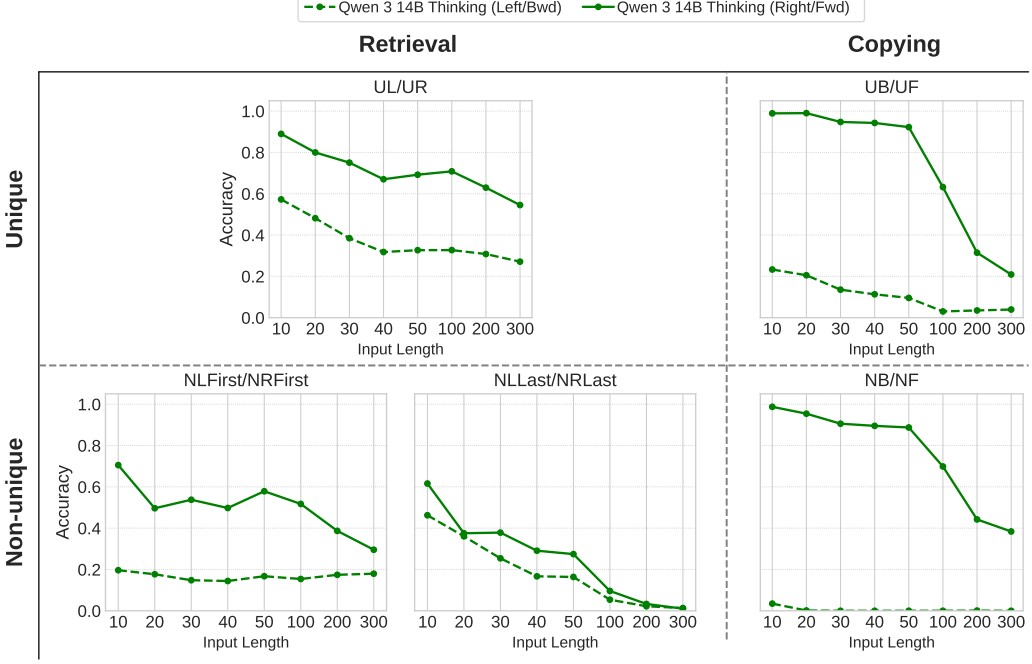

Figure 24: In context accuracy for `Qwen3-14B-Thinking` across all our **word-level** tasks averaged over 3 seeds and prompt variations per task. The same patterns as discussed in Section 4.1 are visible.

tokens per paragraph, aggregated across all random seeds. While different model families employ distinct tokenizers, so their absolute counts vary, yet the relative balance between ambiguous and unambiguous tokens is remarkably consistent. Each paragraph contains roughly 500 tokens, but repetition of sentences (to make the dataset appropriate for Non-unique copying) reduces the number of unique token types to only about 70. However, amongst those, both token types occur frequently in our data and the fact that copying "glitch" can be traced *exclusively* to unambiguous tokens is remarkable.

Table 7: Average per-paragraph bigram counts by model

| Model name | Ambiguous tokens | Unambiguous tokens | Total tokens |
|---|---|---|---|
| `Llama3_8B, Llama3_70B` | 31.97 | 39.65 | 71.62 |
| `Qwen2.5_7B, Qwen2.5_32B` | 32.59 | 42.14 | 74.72 |

**Example of a Hallucinated Chain.** We provide here a truncated real sample (from the total 500 sequence length of the original paragraph) of an input sequence to be copied:

```
... neam que Non etincidunt dolorem tempora magnam.
```

and the corresponding output we get for the same input is:

```
... neam que Non etincidunt dolorem tempora magnam velit neque.
Non etincidunt dolorem tempora magnam.
```

As can be seen, a long chain of hallucination starts at the point `tempora magnam`. However, earlier in the truncated history, there were cases where `magnam` was followed by `velit`.

Our algorithm proceeds with this example as follows. We first tokenize both sequences to get the following lists of tokens in the input and the output. Here, the symbol $\dot{G}$ denotes a leading space for any token:

```
Input:  ['Ġne', 'am', 'que', '.', 'ĠNon', 'Ġet', 'inc', 'idunt', 'Ġdol', 'orem',
'Ġtemp', 'ora', 'Ġmagn', 'am', '.'],

Output: ['Ġne', 'am', 'que', '.', 'ĠNon', 'Ġet', 'inc', 'idunt', 'Ġdol', 'orem',
        'Ġtemp', 'ora', 'Ġmagn', 'am', 'Ġvelit', 'Ġne', 'que', '.',      'ĠNon',
        'Ġet', 'inc', 'idunt', 'Ġdol', 'orem', 'Ġtemp', 'ora', 'Ġmagn', 'am', '.']
```

The alignment we obtain between input and output tokens is:

```
[('match' 0, 0), ('match' 1, 1), ('match' 2, 2), ('match' 3, 3),
 ('match' 4, 4), ('match' 5, 5), ('match' 6, 6), ('match' 7, 7),
 ('match' 8, 8), ('match' 9, 9), ('match' 10, 10), ('match' 11, 11),
 ('match' 12, 12), ('match' 13, 13), ('insert', None, 14), ('insert', None, 15),
 ('insert', None, 16), ('insert', None, 17), ('insert', None, 18),
 ('insert', None, 19), ('insert', None, 20), ('insert', None, 21),
 ('insert', None, 22), ('insert', None, 23), ('insert', None, 24),
 ('insert', None, 25), ('insert', None, 26), ('insert', None, 27)]
```

After grouping the alignment by operation, we get:

```
[('match' (0, 13)), ('insert', [14, 27))]
```

Finally, we analyze the transition index between these groups, which here is the index 13. The token at that index is the word `am`, which appeared earlier in the context and was followed by other tokens and *not* a period (`.`). Since this transition index at `am` is ambiguous, we classify the source of this hallucination chain as *ambiguous*.

### C.5.1 Additional results for longer lengths

We conducted additional experiments with the same Lorem-Ipsum setup described above for a single seed at even longer input lengths (up to 5000 tokens) and report the results in Table 8. We observe the same effects we see at lengths of 500, that is, a lower success rate for ambiguous bigrams than for unambiguous ones. These results on longer inputs align perfectly with and amplify our findings in Section 4.2.

Table 8: Model Performance at Longer (2k/3k/5k) Input Lengths

| Model | Unambiguous | | | Ambiguous | | |
|---|---|---|---|---|---|---|
| | 2k | 3k | 5k | 2k | 3k | 5k |
| llama3-70B-Instruct | 0.99 | 0.99 | 0.99 | 0.96 | 0.94 | 0.76 |
| llama3-8B-Instruct | 0.96 | 0.98 | 0.99 | 0.89 | 0.69 | 0.44 |
| qwen2.5-32B-Instruct | 1.00 | 0.99 | 0.99 | 0.95 | 0.94 | 0.93 |
| qwen2.5-7B-Instruct | 0.99 | 0.99 | 0.99 | 0.75 | 0.43 | 0.19 |

Table 9: Prompt templates for copying Lorem Ipsum styled text. `input` stands for the test paragraph. All templates terminate immediately after the final `<start>` token, leaving the closing tags for the model to complete. The example paragraphs along with their copies are written in full in the templates for few-shot examples (omitted here for brevity)

| Name | Template body |
|---|---|
| ECHO-TAG | RULE : Paragraphs are repeated \*\*exactly\*\* as is, whenever an Echo is seen.

Paragraph:
`<start>` Once upon a time ...  `<end>`
Echo:
`<start>` Once upon a time ...  `<end>`

...
Paragraph:
`<start>` input `<end>`
Echo:
`<start>` |
| VERBATIM-END | Paragraph:

`<start>` input `<end>`

Provide a verbatim copy of the paragraph above exactly as it appears, with no modifications.  Use the tag `<end>` to mark the end of the response.

`<start>` |
| P-TAG-THE_END | Please repeat the following paragraph exactly as it is, without any changes:

input

Put the repeated paragraph between `<paragraph>` and `</paragraph>` tags.  Use the phrase "THE_END" to mark the end of the response.

`<paragraph>` |

## C.6   Prompting: Git Commit Revert vs Cherry-Pick (Section 4.2)

**Dataset generation.**   We synthesize commit logs by creating a randomized seven-character hexadecimal hash, an action/topic message drawn from 12 verbs and 10 subsystems. A *history snippet* lists $\ell$ commits newest,$\rightarrow$ oldest. We export four depths ($\ell = 10, 15, 20, 25$) and three random seeds, yielding $4 \times 3 \times 1,500 = 18,000$ total examples. For each snippet consisting of this generated git commit history (listed newest $\rightarrow$ oldest) we store two reference answers:

- revert: the lines *as shown* (newest,→,oldest), matching the order a developer would pass to `git revert` (thus somewhat similar to forward copying)

- cherrypick: the same lines reversed (oldest,→,newest), the order expected by `git cherry-pick`. (thus somewhat similar to backwards copying)

All files are distributed in `jsonl` format with keys `snippet`, `revert`, and `cherrypick`.

**Prompt variants.** We adopt exactly **two** natural-language templates; one for the cherry-pick orientation, one for the revert orientation. Each prompt describes the task in plain English, includes up to three few-shot examples (hand-crafted, $\ell = 6$), shows the new history under a `=== NEW HISTORY ===` delimiter, and terminates with `=== ANSWER ===`, where the model must emit the ordered commit list only. An example is given below.

**Prompt template examples - Cherry Pick**

```
Below is a part of a Git commit history (newest at the top, oldest at the bottom).

Task
List every commit line that must be cherry-picked onto the release branch.
 Output them **from oldest to newest** (the order 'git cherry-pick' expects).
One complete commit line per output line.
Do **not** output anything else.

{{ few_shot_block }}
=== NEW HISTORY ===
{{ snippet }}
=== ANSWER ===
<start>
```

**Prompt template examples - Revert**

```
Below is part of a Git commit history (newest at the top, oldest at the bottom).

Task
List every commit line that must be reverted to roll the codebase back.
Output the lines **from newest to oldest**.
One complete commit line per output line.
Do **not** output anything else.

{{ few_shot_block }}
=== NEW HISTORY ===
{{ snippet }}
=== ANSWER ===
<start>
```

Because correct output demands global re-ordering rather than token-level copy, we found no benefit in adding separator toggles or mathematical re-statements.

**Model coverage.** Preliminary sweeps revealed that **Qwen2.5 7B/32B** models performed really poorly $\leq 10\%$ on this task, *even at further shorter depths* $\ell = 5$, hence we exclude them from our analysis. Performance degrades further with longer histories, indicating a difficulty with list reversal rather than context window length.

**Additional prompting details for Instruction-tuned model variants.** We keep the prompts very similar to the completion models and mainly change the output formatting instructions (i.e., `Put the answer between <target>, <\target>` tags). Additionally, both `Qwen` and `Llama` family of models require and an additional SYSTEM PROMPT, which we provide as follows:

```
You are a very careful and precise assistant. You always follow the instructions
and solve tasks yourself. You never generate code.
```

With our initial experiments, we observed that unless we specify explicitly, instruction-tuned model variants generated code to solve both copying and retrieval tasks.

# D  Fine-Tuning Details (Section 4.3)

In general, the random seeds only affect the data shuffling and the positional offset during fine-tuning; the rest of the things remain unaffected.

## D.1  Retrieval Tasks - UL/UR

**Examples.**  Some sample inputs with the query token (bolded) and their UL/UR answers

| Input | UL answer | UR answer |
|---|---|---|
| ns0w6u̲p9v8||**u** | 6 | p |
| qyw283̲zd9411w8||**3** | 8 | z |

**Dataset.**  The training/test dataset statistics for the UL/UR tasks are as follows:

| | UL | UR |
|---|---|---|
| # train/val/test samples | 45k/5k/5k | 45k/5k/5k |
| train/val sample lengths [min, max] | [4, 100] | [4, 100] |
| test sample lengths [min, max] | [101, 200] | [101, 200] |

**Hyperparameters.**  The table below contains all the hyperparameters for fine-tuning models on the UL/UR dataset.

| Hyperparameter | Value |
|---|---|
| Model name | GPT-2-XL[12] |
| Tokenization | Character-level |
| Maximum train sequence length | 100 |
| Maximum inference sequence length | 200 |
| # Epochs | 30 |
| Batch Size | 64 |
| Learning rate (AdamW) | 1e-5 |
| Weight decay (AdamW) | 0.01 |
| Warmup ratio | 0.15 |
| Random seeds | 3, 71, 92, 435, 541, 591, 24050, 29214 |
| Maximum gradient norm | 1.0 |

## D.2  Retrieval Tasks - NLFirst/NRFirst/NLLast/NRLast

**Examples.**  Some sample inputs with the query token (char after '||') and their NLFirst/NRFirst/NLLast/NR-Last targets.

| Input | NLFirst | NRFirst | NLLast | NRLast |
|---|---|---|---|---|
| q5o0o8b6v5o3||o | 5 | 0 | 5 | 3 |
| c8r5r5r3r6r0||r | 8 | 5 | 6 | 0 |

**Dataset.**  The training/test dataset statistics for the NLFirst/NRFirst/NLLast/NRLast tasks are as follows:

| | NLFirst/NRFirst/NLLast/NRLast |
|---|---|
| # train/val/test samples | 100k/5k/4.7k |
| train/val sample lengths [min, max] | [4, 100] |
| test sample lengths [min, max] | [101, 200] |

**Hyperparameters.** The table below contains all the hyperparameters for fine-tuning models on the NL-First/NRFirst/NLLast/NRLast dataset.

| Hyperparameter | Value |
|---|---|
| Model name | GPT-2-XL |
| Tokenization | Character-level |
| Maximum train sequence length | 100 |
| Maximum inference sequence length | 200 |
| # Epochs | 15 |
| Batch Size | 64 |
| Learning rate (AdamW) | 1e-5 |
| Weight decay (AdamW) | 0.01 |
| Warmup ratio | 0.15 |
| Random seeds | 3, 47, 71, 100 |
| Maximum gradient norm | 1.0 |

## D.3 Copying Tasks - UF/UB/NF/NB

**Examples.** Some sample inputs and their UF/UB/NF/NB targets.

| Input | Config | Target |
|---|---|---|
| Syb5DEHihO> | UF | Syb5DEHihO |
| Syb5DEHihO> | UB | OhiHED5byS |
| LnvTs1qgMt> | UF | LnvTs1qgMt |
| LnvTs1qgMt> | UB | tMgq1sTvnL |
| 9975813713> | NF | 9975813713 |
| 9975813713> | NB | 3173185799 |
| 525671167> | NF | 525671167 |
| 525671167> | NB | 761176525 |

**Dataset.** The training/test dataset statistics for the UF/UB/NF/NB tasks are as follows:

| | UF/UB/NF/NB |
|---|---|
| # train/val/test samples | 50k/5k/5k |
| train/val sample lengths [min, max] | [4, 100] |
| test sample lengths [min, max] | [101, 200] |

The train/val sample lengths are considered without the delimiter but the combined input/target length is 100, not just the input length.

**Hyperparameters.** The table below contains all the hyperparameters for fine-tuning models on the UF/UB/NF/NB dataset.

| Hyperparameter | Value |
|---|---|
| Model name | GPT-2-XL |
| Tokenization | Character-level |
| Maximum train sequence length | 100 |
| Maximum inference sequence length | 200 |
| # Epochs | 15 |
| Batch Size | 64 |
| Learning rate (AdamW) | 1e-5 |
| Weight decay (AdamW) | 0.01 |
| Warmup ratio | 0.15 |
| Random seeds | 3, 33, 71, 77, 91 |
| Maximum gradient norm | 1.0 |

### D.4  Results

We provide the training loss curves for each fine-tuning task in Figure 25.

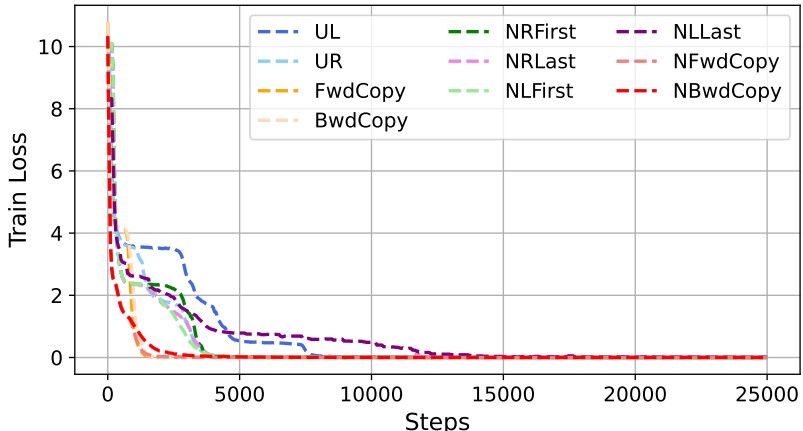

Figure 25: Averaged training loss curve for `GPT-2-XL` model across all our tasks. For all the tasks, we achieve 0 training loss. Yet, the generalization behavior varies in line with theoretical predictions (Figure 6).

## E  Further Details on Mechanistic Interpretability Analysis (Section 4.4)

**Dataset.**  We randomly generate 100 strings containing letters of the English alphabet in lowercase. To prompt base models to perform the tasks, we add 10 examples before the actual query for few-shot learning. We tokenize each symbol individually, adding a preceding white space to each letter to get a more natural tokenization. We add a BOS token in the beginning, separate few-shot examples with a dot, and separate the input and output of each example with a comma. Fine-tuned models use the same dataset, except that tokenization and the choice of separators are inherited from the fine-tuning setup, and they also have no BOS token and do not require few-shot examples.

**Patching details.**  To remove an induction head, we use path patching methodology, widely adopted in the literature [Hanna et al., 2023, Wang et al., 2022]. An induction head can be viewed as a path consisting of two edges connecting different positions inside an attention head: one edge connecting two adjacent positions in the input string, and one edge connecting the second of these with a position in the target string. We remove paths inside attention heads following Bakalova et al. [2025]. Removing all induction heads is thus done in two forward passes:

- In all heads, replace the V activation of a token at position $P$ in the input string with zero whenever it is queried by the token in position $P + 1$ in the input string. In the same forward pass, for each head, save K and V activations at position $P + 1$ in the input string. *Intuitively, this intervention prepares K and V activations at each position that carry no information about the preceding token.*
- In a second forward pass, in each head, replace the K activation of the token in position $P + 1$ in the *input* string with the activation saved in the previous step whenever it is queried by a token in position $P$ in the *target* string. *Intuitively, this intervention ensures that a head cannot attend to a position on the basis of information about the token immediately preceding the key position. That is precisely the behavior of induction heads [Olsson et al., 2022], i.e., induction heads are removed by this intervention.*

When removing anti-induction heads, the first step is the same, but the second step is different: In each heads, we replace the V activation of the token in position $P + 1$ in the *input* string with the activation saved in the first step whenever it is queried by a token in position $P + 1$ in the *target* string. *Intuitively, this intervention ensures that an attention head attending to a prior occurrence of the same token cannot retrieve the immediately preceding token.*

The above description applies to Unique Forward Copy; in Unique Backward Copy, the position indices in the target string are appropriately reversed.

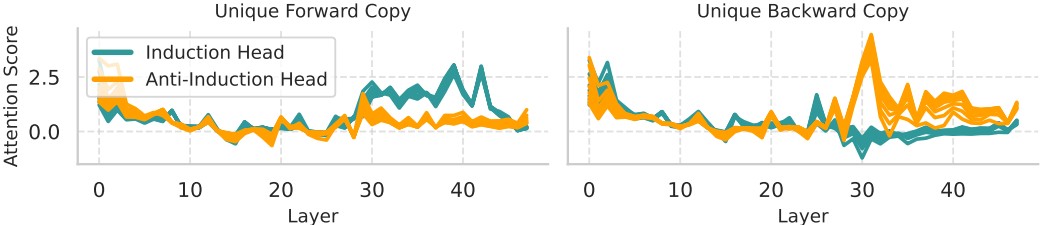

Figure 26: Difference in sum of attention scores per layer between fine-tuned and base models. The higher the difference, the more amplified is this head in this layer during fine-tuning.

**Layer localization.** In fine-tuning, useful heads may migrate upward in depth (Figure 26). In fine-tuned models, induction (resp. anti-induction) strength peaks in the top-third layers for forward (resp. backward) copy. Pretrained models exhibit the same pattern but with induction heads situated later than anti-induction heads, mirroring their superior robustness.

**Relation to Retrieval Heads.** Recent work by Wu et al. [2025] introduced the notion of a *retrieval heads*. The experiments to find these retrieval heads in Wu et al. [2025] focus primarily on zero-shot settings compared to the typical few shot settings for induction head studies [Elhage et al., 2021, Olsson et al., 2022, Song et al., 2025, Crosbie and Shutova, 2025] including our setup. These retrieval heads are found by looking at tokens being retrieved from Needle in a Haystack like test beds [Liu et al., 2024b]. There is a needle, or a short answer span that needs to be copied in the output given a query. These answer spans / needles consist of tokens that are sufficiently unique in the haystack (context) and thus cannot be figured out on the basis of semantics alone. A head is called a retrieval head if it pays maximal attention to tokens being copied from a needle more number of times than any other head. For example, if the number of tokens to be copied from a needle is 10, and the maximal retrieval score that was calculated came out to be 0.7, that would imply that the head in question paid the maximal attention to the 7 out of the 10 tokens while they were being copied. Thus even though Retrieval heads are highly connected to the way we calculate our induction and anti-induction circuits, the exact methodology is different. However retrieval heads should have a strong connection with the induction heads, when they perform forward copying, and the anti-induction heads when they perform reverse copying. The exact correlation however is left for future work.

## F Compute Resources

We run all the prompting experiments using 4B/7B/8B parameter models on a single H100 GPU and 14B/32B/70B parameter models on 4xH100 GPUs with full precision without any quantization. Our fine-tuning experiments also use a single H100 GPU for all the experiments. All of our experiments should be reproducible with approximately 2500 H100 GPUh.

