# OpenReview forum: "Born a Transformer -- Always a Transformer? On the Effect of Pretraining on Architectural Abilities"
_NeurIPS.cc/2025/Conference — NeurIPS 2025 poster_

### Official Review · Reviewer_akQM · 2025-06-25

**Clarity:** 4
**Significance:** 2
**Originality:** 3
**Rating:** 4
**Confidence:** 4

**Summary:**

The goal of this paper is to quantify whether and how the length generalization limitations predicted by theory persist in large pretrained transformers, particularly in tasks involving retrieval and copying. The authors propose variants of forward/backward copying and left/right retrieval tasks, and assess the models' length generalization capabilities as predicted by C-RASP. They also empirically validate the accuracy of C-RASP predictions by finetuning small models on each task. Subsequently, they evaluate pretrained models in-context on these tasks and reveal a stark contradiction between empirical performance and theoretical predictions. Additionally, they observe that pretrained models perform significantly better on the forward versions of the tasks compared to their backward counterparts. This discrepancy is later attributed to the relative strength of induction circuits over anti-induction circuits, likely shaped by patterns in the pretraining data.

**Questions:**

1) In addition to my earlier point in the weaknesses section: if the goal of Section 4.1 is to cross-check the C-RASP length generalization predictions on pretrained language models, it’s important to note that the C-RASP framework assumes learned absolute positional embeddings. However, the models evaluated in this section use RoPE. This discrepancy is not clearly acknowledged or discussed in the paper, and it likely introduces a confounding factor into the evaluation.

2) The finding in Section 4.2 and Figure 4 closely resembles the results of Jelassi et al. [2024], who demonstrated that transformers rely on  bigram lookup mechanisms for copying. It would be helpful to more explicitly acknowledge (or possibly differentiate) this.

3) The discussion on “token-level alignments” (lines 217–222) and the “Git Commit History” experiment were difficult to follow and could benefit from clearer explanations.

4) Is there a particular reason the performance of the GPT-2 model prior to fine-tuning isn't reported? Including this baseline would help quantify the relative gains achieved through fine-tuning on each task.

5) The task names (e.g., UL vs. UB) are somewhat confusing when reading the text. As a suggestion, perhaps adding a prefix, like Copy-U-L or Retrieve-U-B could improve readability and help readers quickly identify tasks.

**Ethical Concerns:**

["NO or VERY MINOR ethics concerns only"]

**Final Justification:**

The paper is clear and well-presented, with informative results and thorough experiments. Its main motivation is to assess how transformers' theoretical limitations in length generalization manifest in large-scale pretrained models. However, this question may not be fully addressed, as the tested sequences are relatively short. Instead, the authors observe a left–right asymmetry in models' recall capabilities that is likely influenced by the training data. While this finding is interesting, another minor concern is the significance of the contribution for NeurIPS.

**Limitations:**

yes

**Quality:**

3

**Strengths And Weaknesses:**

**Strengths:**

The paper is well-organized, well-written, and easy to follow. The scope is clearly defined, the tasks are thoroughly explained with helpful schematics, and the use of paragraph titles and takeaway boxes summarizes the main points of each section.

The experiments are clearly presented and rigorously documented. I also appreciated the variety of in-context prompting variants discussed in the appendix. The main results, particularly those in Figures 3 and 4, are highly informative with clear practical implications.

**Weaknesses:**

My main concern is that one of the key motivations of the paper is to examine whether theoretical limitations on _length generalization_ actually impact the performance of large pretrained models in practice. However, I believe this question is most meaningful when tested on relatively long sequences, whereas the experiments in Section 4.1 focus on rather short ones.

Model performance mainly depends on whether the sequence lengths used during testing were also present during training. As a result, performance (especially on shorter sequences that are more common in training data) may reflect the influence of the training distribution more than any fundamental architectural limitations.

The observation in Section 4.2, where models fail to copy because of ambiguous tokens, aligns with the theory prediction and is performed on longer sequences. This also supports the idea that the sequences used in Section 4.1 may have been too short to properly test the predictions of the length generalization theory.

Therefore, the observed gap between theoretical predictions and empirical performance might be more attributable to the nature of the training data than to the scale of the models or the web-scale data they were trained on, as suggested in the introduction (lines 37–41). I believe this point should be clarified more.

---

> ### Author Rebuttal · Authors · 2025-07-31
>
> We thank the reviewer for their detailed feedback regarding clarifications on theoretically predicted generalization behavior, experimental setups, and writing improvements. We intend to address the above mentioned weaknesses and questions in separate sections below.
>
>
> #### Response to Weaknesses
> > My main concern is that one of the key motivations of the paper is to examine whether theoretical limitations on length generalization actually impact the performance of large pretrained models in practice. However, I believe this question is most meaningful when tested on relatively long sequences, whereas the experiments in Section 4.1 focus on rather short ones.
> 1. Indeed, we acknowledge that the sequence limit of 50 tokens in Section 4.1 does not exhaust the potential real world sequence lengths. The important point in Section 4.1 is that the key failures already appear at shorter lengths, which are potentially more frequent in the training data and particularly practically relevant. As the reviewer mentioned, we also test longer sequences up to 500 tokens on the Lorem-Ipsum-style copying task in the Section 4.2.
>
>
> > Model performance mainly depends on whether the sequence lengths used during testing were also present during training. As a result, performance (especially on shorter sequences that are more common in training data) may reflect the influence of the training distribution more than any fundamental architectural limitations.
>
> > The observation in Section 4.2, where models fail to copy because of ambiguous tokens, aligns with the theory prediction and is performed on longer sequences. This also supports the idea that the sequences used in Section 4.1 may have been too short to properly test the predictions of the length generalization theory.
>
> > Therefore, the observed gap between theoretical predictions and empirical performance might be more attributable to the nature of the training data than to the scale of the models or the web-scale data they were trained on, as suggested in the introduction (lines 37–41). I believe this point should be clarified more.
>
> 2. While we agree with the reviewer that the influence of the training distribution does impact performance in many ways, that is precisely why we found our results exciting: our task setups allowed us to disentangle the effects of pretraining from architectural limitations. Through our experiments in Section 4.1, we were already able to observe the effects of pretraining — certain abilities were selectively amplified more than others, despite there being little to no theoretical difference between them.
> Since we were able to shortlist which abilities the training distribution amplifies the most, we moved on to the experiments in Section 4.2. We are glad the reviewer agrees that the failure modes observed in Section 4.2 indeed align with theoretical predictions.
> It was only in Section 4.3, through our fine-tuning experiments and mechanistic insights, that we were able to conclude that the directional asymmetry (e.g. left/right or forward/backward) was due to pretraining, while certain other failures (e.g. in retrieval models, or repeated copying) could be attributed to architectural limitations. In these experiments, we tested our length generalization setup by training models on shorter sequences and evaluating them on longer ones. We were able to eliminate failures caused by pretraining, but the effects of architectural limitations remained.
> This helped us finally disentangle the effects of pretraining from those of architectural limitations. As suggested by the reviewer, we will make this point clearer in the main text of the paper.
>
>
> #### Response to Questions
> > In addition to my earlier point in the weaknesses section: if the goal of Section 4.1 is to cross-check the C-RASP length generalization predictions on pretrained language models, it’s important to note that the C-RASP framework assumes learned absolute positional embeddings. However, the models evaluated in this section use RoPE. This discrepancy is not clearly acknowledged or discussed in the paper, and it likely introduces a confounding factor into the evaluation.
>
> 1. Indeed, the C-RASP[pos] framework assumes a Transformer architecture using APE. In the footnote of the Section 3 we note that while Huang et al. results rely on this assumption, we find that empirically these results extend to the RoPE setting, which are more extensively discussed in the Appendix B, providing the from-scratch results for both APE and RoPE Transformers. However, we agree to make this note more explicit in the body of the paper.
>
> > The finding in Section 4.2 and Figure 4 closely resembles the results of Jelassi et al. [2024], who demonstrated that transformers rely on bigram lookup mechanisms for copying. It would be helpful to more explicitly acknowledge (or possibly differentiate) this.
>
> 2. Although we do cite the results of Jelassi et al. [2024], we agree that we could make this discussion more explicit, and we promise to do so.
>
> > The discussion on “token-level alignments” (lines 217–222) and the “Git Commit History” experiment were difficult to follow and could benefit from clearer explanations.
>
> 3. We acknowledge that the token-level alignment and the Git Commit task can be described more clearly. We had hoped that the additional details and example traces provided in Appendix C.3 would help clarify our method. Nonetheless, we will make a concerted effort to further improve the explanations in the main body of the paper.
>
> > Is there a particular reason the performance of the GPT-2 model prior to fine-tuning isn't reported? Including this baseline would help quantify the relative gains achieved through fine-tuning on each task.
>
> 4. We found that the baseline performance across tasks was at a chance level (i.e. the baseline was 0), so, we have not included those in-context results. We will consider making these results explicit in Figure 6.
>
> > The task names (e.g., UL vs. UB) are somewhat confusing when reading the text. As a suggestion, perhaps adding a prefix, like Copy-U-L or Retrieve-U-B could improve readability and help readers quickly identify tasks.
>
> 5. Thank you for your suggestions. We will carefully consider ways to improve readability of the paper, including potential task names clarifications.

---

> > ### Author Response · Authors · 2025-08-05
> >
> > We again thank the reviewer for their feedback, and we wonder if the reviewer believes that our response fully answers their questions and concerns. We would appreciate if the reviewer could provide us with more feedback on the matter.

---

> > ### Comment · Reviewer_akQM · 2025-08-06
> >
> > I would like to thank the authors for their detailed response and clarification.
> >
> > I have no further questions or feedback, except for one minor point: I agree that the footnote in Section 3 regarding the experiments on RoPE vs. APE should be highlighted more clearly, ideally with a pointer in the main text. However, I am somewhat skeptical of the claim that "APE results extend to RoPE empirically." As stated in Appendix B, the model does not achieve perfect length generalization with RoPE. Instead, the authors train on a surrogate format to demonstrate equivalent results.

---

> > > ### Author Response · Authors · 2025-08-07
> > >
> > > We thank the reviewer for their comment and promise that we would make these and above mentioned details more explicit in the final paper.

---

### Official Review · Reviewer_gWjY · 2025-07-01

**Clarity:** 3
**Significance:** 3
**Originality:** 3
**Rating:** 5
**Confidence:** 3

**Summary:**

The paper investigates whether fundamental length-generalization limits of small transformers carry over to large pretrained LLMs. Using a family of synthetic retrieval and copying tasks formalized in the C-RASP[pos] framework, the authors derive theoretical expressivity guarantees. They then measure in-context performance across multiple LLMs and uncover a consistent right-forward versus left-backward asymmetry, trace this mechanistically down to induction versus anti-induction head prevalence, and show that supervised fine-tuning rebalances these circuits to restore theory-aligned generalization.

**Questions:**

- For the git cherry picking task, some performance drop might be due to the LLM's unfamiliarity with this command. Would be good to have a more straight forward task.
- Generality of this paper would benefit from evaluating on SOTA closed-source systems.

**Ethical Concerns:**

["NO or VERY MINOR ethics concerns only"]

**Final Justification:**

The authors conducted a series of theory-guided experiments for understanding retrieval and copying tasks. All of my questions have been addressed.

**Limitations:**

yes

**Quality:**

3

**Strengths And Weaknesses:**

**Strength:** The paper combines solid theory with extensive empirical validation on both synthetic and naturalistic tasks, connecting C-RASP[pos] expressivity to observed LLM behavior and mechanistically attributing asymmetry to circuit strength.

**Weakness:** Real-world evaluations remain limited to specific tasks at moderate context lengths. The paper can also benefit from some more discussions on the initial asymmetry.

---

> ### Author Rebuttal · Authors · 2025-07-31
>
> We are thankful to the reviewer for their comments and aim to address the concerns regarding testing lengths, asymmetry discussion, and evaluation choices below.
>
>
> #### Response to Weakness:
>
> > Real-world evaluations remain limited to specific tasks at moderate context lengths.
> 1. While we acknowledge that real-world applications involve processing of larger text sequences, we believe it to be important and interesting that the mentioned effects can already be observed at smaller lengths. Additionally, Section 4.2 does extend our experiments to larger context lengths, showcasing a more naturalistic test bed.
>
>
> #### Response to Questions:
>
> > The paper can also benefit from some more discussions on the initial asymmetry.
>
> 2. We are unsure what the reviewer means by *initial asymmetry*. If this refers to the from-scratch behavior in comparison to the behavior of pretrained LLMs, we report those results in Appendix B. Otherwise, we would be grateful if the reviewer could clarify this in the discussion.
>
>
> > For the git cherry picking task, some performance drop might be due to the LLM's unfamiliarity with this command. Would be good to have a more straight forward task.
>
> 3. While we acknowledge that the model's familiarity with the *git cherry pick* command might affect the performance, we provided an explanation of the task which uses the same terms as the terms mentioned for the *git revert* task, and also provided three few-shot examples in the prompt to further explain what the task is, which taken in conjuction should alleviate the effect of token unfamiliarity, as we observed in our experiments in section 4.1 as well. Despite this, we still find a gap in performance between these tasks, matching the theoretical predictions.
>
> > Generality of this paper would benefit from evaluating on SOTA closed-source systems.
>
> 4. The focus of our paper is to understand the role of architectural limitations and biases from the pretraining data in shaping LLMs' abilities. These details are not known for closed-source systems -- thus, evaluation of the same setup in closed-source setting, even if results are similar, cannot be directly attributed to the model's architecture or pre-training regimen. We thus believe that open systems are most relevant to testing our research questions.

---

> > ### Comment · Reviewer_gWjY · 2025-08-04
> >
> > I'd like to thank authors for addressing my questions. I have raised my score to 5.

---

### Official Review · Reviewer_KxkG · 2025-07-02

**Clarity:** 3
**Significance:** 3
**Originality:** 3
**Rating:** 4
**Confidence:** 4

**Summary:**

In the present paper, the authors investigate whether Transformer architectures might display unexpected length-generalization limitations compared to those predicted theoretically (e.g., C-RASP[pos]), after large-scale pretraining. The authors focus on copying and retrieval tasks, comparing model performance on “rightward” (induction) versus “leftward” (anti-induction) variants, and highlighting a robust right-hand bias in pretrained LLMs (LLaMA-3, Qwen2.5), even though theory predicts symmetric behavior. This asymmetry is linked to induction vs. anti-induction heads, and it disappears with targeted fine-tuning. However, some additional experiments show how “glitches” arise in non-unique copying due to ambiguous transitions, and these cannot be easily removed in a fine-tuning stage. The work concludes that pretraining amplifies certain capabilities but does not eliminate architecture-induced biases.

**Questions:**

1 - Isn’t a rightward bias natural given the autoregressive training? Why is this presented as an architectural failure rather than an expected outcome?

2 - Is there a connection between the glitch phenomenon and the induction/anti-induction head asymmetry? Do the authors believe that predicting multiple tokens at the same time (e.g. with diffusion-based methods) could remove this bias?

3 - Does the existence of a C-RASP[pos] program prove that a task will generalize under all training regimes and model types?

4 - Why was GPT-2 used for fine-tuning instead of LLaMA/Qwen? Do the authors expect the results to hold for modern LLMs? Can the authors provide a baseline for comparison showing the bias of GPT2 before fine-tuning?

**Ethical Concerns:**

["NO or VERY MINOR ethics concerns only"]

**Final Justification:**

I believe this work shows an interesting pre-training artifact in LLMs, that should be accounted for in specific applications. Overall, the presence in the main of two parts that are not well-connected, and the lack of rigor in the extension of the proposed theory to the settings where the experiments are run limit somewhat the readability and impact of this work. For these reasons, I will maintain my initial score.

**Limitations:**

Discussed.

**Quality:**

3

**Strengths And Weaknesses:**

**Strengths**
- The asymmetry in performance is consistent and well-demonstrated across tasks and models.
- The paper convincingly links asymmetry to the prevalence of induction vs. anti-induction heads, across models, instruction vs. non-instruction tuning, and multiple prompt templates.
- The authors also provide a formal lens to understand the phenomenology, using the C-RASP[pos] framework.
- The authors convincingly discuss that these subtle biases can manifest in practical contexts (Git workflows) or copying long neutral text.

**Weaknesses**
- The paper does not fully address whether this “bias” is surprising, especially given that Transformers are trained autoregressively (left-to-right).
- While reading the paper, it remains ambiguous whether the glitches in non-unique copying are related to the directional asymmetry or are a separate phenomenon entirely. In this case, it is not completely clear what this behavior adds to the "unexpected bias from pretraining" picture.
- To a non-expert reader, it is unclear if the presence of a C-RASP[pos] program implies a necessary and sufficient condition for generalization, or just an empirical trend.
- The explanation/analysis of the induction head mechanism appears to be slightly rushed if one does not assume prior familiarity with mechanistic interpretability.
-  The fine-tuning experiments (showing that the bias can be removed) are conducted on GPT-2 model rather than the same models (LLaMA, Qwen) used earlier, making comparisons less direct.

---

> ### Author Rebuttal · Authors · 2025-07-31
>
> We thank the reviewer for their feedback and suggestions on clarifications of explanations. We would like to jointly respond to question and weaknesses below.
>
> > from Weaknesses: The paper does not fully address whether this “bias” is surprising, especially given that Transformers are trained autoregressively (left-to-right).
>
> > from Questions: Isn’t a rightward bias natural given the autoregressive training? Why is this presented as an architectural failure rather than an expected outcome?
>
> 1. We would like to clarify that we consider the rightward bias to be *an artifact of pretraining*, and *not an architectual failure*. Indeed, we agree with the reviewer this it is a natural artifact of autoregressive training on language. We consider it to be especially interesting as the theory predicts equal difficulty of both directions for the architecture for precisely autoregressive generation of output by Transformers. So, despite the bias being intuitive, we consider finding a more precise reason to explain it to be worthwhile. We believe that explaining this bias mechanistically using induction and anti-induction circuits gives an important empirical insight relevant for downstream applications.
>
>
> > from Weaknesses: While reading the paper, it remains ambiguous whether the glitches in non-unique copying are related to the directional asymmetry or are a separate phenomenon entirely. In this case, it is not completely clear what this behavior adds to the "unexpected bias from pretraining" picture.
>
> > from Questions: Is there a connection between the glitch phenomenon and the induction/anti-induction head asymmetry? Do the authors believe that predicting multiple tokens at the same time (e.g. with diffusion-based methods) could remove this bias?
>
> 2. The copying glitches and the left-right asymmetry are separate phenomena. The copying glitches are tightly related to the inherent architectural difficulty of non-unique copying and retrieval. In contrast, the left-right asymmetry is a bias arising from pre-training. Theory describes the existence of the copying glitches as inherent to Transformer architecture, while asymmetry is removable by fine-tuning. We believe that fine-tuning the model to remove the above mentioned asymmetry is already a good way to address the issue in more practical settings. Non-autoregressive methods lie beyond the scope of our paper, which focuses on Transformers. Confirming the existence or absence of both the glitches and the left-right asymmetry in such architectures is an interesting future work direction that might be a promising solution on par with fine-tuning.
>
> > from Weaknesses: To a non-expert reader, it is unclear if the presence of a C-RASP[pos] program implies a necessary and sufficient condition for generalization, or just an empirical trend.
>
> > from Questions: Does the existence of a C-RASP[pos] program prove that a task will generalize under all training regimes and model types?
>
> 3. Thanks for this question. We would like to point out that, in Section 3, we sketch the theoretical length generalization conditions, including required positional encodings and training procedures. To provide more detail, we will expand this discussion in Section 3 along the following lines: *the theoretical length generalization guarantee for C-RASP[pos] by Huang et al. 2024 applies to APE Transformers under an idealized training procedure, where all training data up to some maximum input length N are available and the exact minimizer of a regularized loss is found. Under these specific conditions, the existence of a C-RASP[pos] program for that task guarantees length generalization to larger lengths. While theory has only been worked out for this specific setup, empirically we observe converging results across different PE types and realistic SGD-based training (as shown in Huang et al. 2024 as well as in the experiments in Appendix B).* In the future, we would be excited to see the extensions of the existing theory to more PE types and more practical regimes.
>
> > from Weaknesses: The fine-tuning experiments (showing that the bias can be removed) are conducted on GPT-2 model rather than the same models (LLaMA, Qwen) used earlier, making comparisons less direct.
>
> > from Questions: Why was GPT-2 used for fine-tuning instead of LLaMA/Qwen? Do the authors expect the results to hold for modern LLMs? Can the authors provide a baseline for comparison showing the bias of GPT2 before fine-tuning?
>
> 4. Regarding the baseline of GPT-2 before finetuning: In experiments, we found that the in-context performance of GPT-2 was at chance level (i.e. 0), hence, we did not include those results in the paper. We will consider making these results explicit in Figure 6. Regarding the choice of GPT-2: Due to computational budget constraints as well as their usage of RoPE, we did not fine-tune larger, 7-8B and 70B models used in the Section 4.1 at the time of the submission.
>
> > from Weaknesses (no corresponding Question): The explanation/analysis of the induction head mechanism appears to be slightly rushed if one does not assume prior familiarity with mechanistic interpretability.
>
> 5. Thank you for your feedback. We will strive to improve this.

---

> > ### Comment · Reviewer_KxkG · 2025-08-05
> > **Comment**
> >
> > I would like to thank the authors for their thoughtful responses. I have some quick follow-ups:
> >
> > > On point 1:
> >
> > Maybe stating this more openly in the main would be useful, even if the authors consider this to be obvious. As a reader, I wasn't sure I was missing something when I read about this effect.
> >
> > > On point 2:
> >
> > It remains unclear how the two phenomenologies tie in with the story of the paper. For example, in the abstract, the focus is fully on the asymmetry effect. While reading the paper, one expects there to be a closer link when the second effect is presented. I believe there needs to be some revision in the presentation fo fix this issue.
> >
> > > On point 3:
> >
> > So do the guarantees on length-generizability apply e.g. to the specific setting considered in this study? Also this aspect should be made clearer (especially if the theory does not fully extend to the considered settings).
> >
> > > On point 4:
> >
> > I believe the authors should at least mention this result, since the model switch is not addressed in the current version.

---

> > > ### Author Response · Authors · 2025-08-05
> > >
> > > We appreciate the reviewer’s feedback and will incorporate all the suggested revisions to improve the clarity and quality of the writing. We are pleased to note that the reviewer has no concerns regarding the experimental setup or the scope of the experimentation. To confirm, if the reviewer’s primary concerns relate to the presentation only and not the methodology, we will strive to address all the constructive feedback provided by the reviewer accordingly.
> > >
> > > 1. Thank you for your feedback, we will make this more explicit in the final version of the paper.
> > >
> > > 2. The phenomena of copying glitches and left-right asymmetry, being separate, illustrate two distinct phenomena from the theoretical point of view: one (copying glitches) follows from architectural limitations, and another one (left-right asymmetry) is not theoretically inherent to the architecture, but introduced by the pretraining data. We will explain this more explicitly in the introduction.
> > >
> > > 3. The length generalization setting described in Section 3 most closely matches the fine-tuning setup of the GPT-2 model, but applies to an idealized training method (not SGD). Again, empirically we find that this extends to other PE types and more realistic SGD-based training. As mentioned above, we will expand this discussion in Section 3 to further clarify this point.
> > >
> > > 4. Thank you for your comment, we will add this explicitly in the revised version.

---

### Official Review · Reviewer_JBEt · 2025-07-05

**Clarity:** 2
**Significance:** 2
**Originality:** 2
**Rating:** 4
**Confidence:** 3

**Summary:**

The paper studies whether large-scale pretraining helps remove certain limits of the transformer architecture, like the length generalization limit. The benchmark task being used for evaluating such capabilities is the "retrieval and copy" testbed. The major finding is that pertaining helps boost certain capabilities, such as achieving near perfect performance on copying, but it shows less strong performance with retrieval. It is also hypothesized that pretraining introduces bias like favoring forward/right induction over the reverse. Such asymmetries can be alleviated in finetuning as inductions in both directions can be amplified. The conclusion is that pretraining can't fundamentally remove the essential inductive bias of the transformer architecture.

**Questions:**

Please refer to the "strengths and weaknesses" above

**Ethical Concerns:**

["NO or VERY MINOR ethics concerns only"]

**Final Justification:**

Supplemented experiments on longer seq_len making the results more convincing

**Limitations:**

yes

**Quality:**

3

**Strengths And Weaknesses:**

Strengths
- The paper tackles a fundamental problem. It is crucial to understand the limitations of the transformer architecture e.g. which tasks are more difficult than the rest and which inductive biases were accountable for them. More importantly how can we leverage those learnings to improve the performance of pretraining and downstream tasks.
- The choice of the copy and retrieval task looks like a simple yet fundamental one and helps test the directional inductive biases

Weaknesses
- A seq_len limit of 50 tokens is too low for any real world application which often takes at least thousands of tokens. A more practical setup is needed to understand the full picture.
- Fig 3 is not enough to conclude that the asymmetry was introduce by pretraining. A comparison against models without pretraining is needed for attribution
- The connection between the "copy and retrieval" task with prestraining ppl or downstream tasks performance is unclear. More specifically how does the learning from the model's behavior on this task generalize to other tasks and if any constructive guidances can be distilled from it. The significance of the findings needs more elaboration
- The paper can benefit from better presentation and story telling. e.g. a contribution summary at the end of the intro section, as well as  the reason to choose copy and retrieval as a representative task would be helpful.

---

> ### Author Rebuttal · Authors · 2025-07-31
>
> We thank the reviewer for their suggestions and comments about the paper. As there were no explicit questions from the reviewer, we address the potential weaknesses below:
>
> > A seq_len limit of 50 tokens is too low for any real world application which often takes at least thousands of tokens. A more practical setup is needed to understand the full picture.
>
> - The seq_len limit of 50 tokens applies specifically to Section 4.1. We'd like to clarify that Section 4.1 is not about practical setups, but the point here is to cleanly check theoretical predictions about uniqueness and left/right, forward/backward (a)symmetries. We swept lengths from 10 to 50 tokens across multiple model sizes and families and observed the same asymmetry at every length. Because the pattern was already stable in this range, we chose 50 as a convenient cutoff. We note that a positive result (i.e., no difference across task setups) at short lengths would have motivated further scaling, whereas a negative result (a clear pattern towards asymmetry in one direction over the other) on such basic tasks is itself striking.  Based on our observations in Section 4.1, we were then motivated to examine longer lengths, which led to the more practical experiments described in Section 4.2. By extending the testing lengths and making the tasks more realistic—such as copying Lorem Ipsum–style paragraphs and performing git revert and cherry-pick on a list of commits—we were able to replicate the findings from Section 4.1. These larger-scale experiments again demonstrated the same asymmetries between left/right or forward/backward directions (as in the git commits case), as well as the relative ease of unique or unambiguous tasks compared to non-unique or ambiguous ones (as in the copying of lorem ipsum paragraphs).
>
> - To further strengthen our point, we have conducted some additional experiments with the Lorem-Ipsum setup at even longer scales (up to 2000 tokens) and report the results in the table below for 2 of our models. Once again, **we do see the same effects we see at lengths of 500, i.e., a lower success rate for ambiguous bigrams than for unambiguous ones.** Running these models for longer and longer lengths is computationally expensive, and as we consistently see these effects at lengths up to 2000, we have concluded our experiments here. If copying had been perfect at such lengths, it would have warranted a deeper analysis at longer lengths. However, since we already observe failures in copying across models and settings at ~500 tokens, we consider it a noteworthy finding. We find this especially interesting as previous work has shown that models perform perfect accuracy while copying ordinary text (with semantic cues) [1] at thousands of tokens length, whereas our Lorem-Ipsum setup demonstrates failures on copying text without semantic cues at much more moderately sized lengths.
>     | Model         | Unambiguous | Ambiguous  |
>     |---------------|-------------|------------|
>     | llama3-70B-Instruct    | 0.99  | 0.96
>     | llama3-8B-Instruct      | 0.96  | 0.89
>     | qwen2.5-32B-Instruct | 1.0 | 0.95
>     | qwen2.5-7B-Instruct | 0.99 | 0.75
>
>
> - Overall, we appreciate the reviewer's comment and will report these results at higher lengths more explicitly in the paper.
>
> > Fig 3 is not enough to conclude that the asymmetry was introduce by pretraining. A comparison against models without pretraining is needed for attribution
>
>
> 2. We would like to clarify our argument that the asymmetry is introduced by pretraining: We agree that the left-right asymmetry is shown most clearly in comparison against models without pretraining, and we would like to point out the from-scratch results in Appendix B. These show that no left-right asymmetry is present when training from scratch. Further important support for the asymmetry as a pretraining artifact is given in Section 4.4, where we give a mechanistic explanation to both fine-tuned and pretrained behavior, and attribute the asymmetry to an imbalance between induction and anti-induction circuits in pretrained models. Taken together, we agree with the Reviewer that the point is not made by Figure 3 -- rather, it follows from comparison to from-scratch models and from mechanistic interventions. We will revise to ensure this argument is made transparent.
>
> > The connection between the "copy and retrieval" task with prestraining ppl or downstream tasks performance is unclear. More specifically how does the learning from the model's behavior on this task generalize to other tasks and if any constructive guidances can be distilled from it. The significance of the findings needs more elaboration
>
> 3. We believe that copy and retrieval tasks are indeed fundamental to reasoning, as solving subproblems in the reasoning chain often involves some sort of retrieval and copying of the input [2]. However, we agree that this can be made more explicit, and we will add a more detailed discussion in the Appendix.
>
> > The paper can benefit from better presentation and story telling. e.g. a contribution summary at the end of the intro section, as well as the reason to choose copy and retrieval as a representative task would be helpful.
>
> 4. Thank you for your suggestions, we will incorporate them into the paper.
>
>
> [1] [Forgetting Curve: A Reliable Method for Evaluating Memorization Capability for Long-Context Models](https://aclanthology.org/2024.emnlp-main.269/) (Liu et al., EMNLP 2024)
> [2] [Query-Focused Retrieval Heads Improve
> Long-Context Reasoning and Re-ranking](https://openreview.net/forum?id=KG9IxJ9ZAL) (Zhang et al., 2025.)

---

> > ### Author Response · Authors · 2025-08-05
> >
> > Thank you once again for your thoughtful review and feedback on our paper.
> >
> > We are writing this brief follow-up comment as we have now completed the even longer-length (i.e., for 3000 and 5000 tokens) experiments than we mentioned in our rebuttal (i.e., 2000 tokens). We are fully aware that the official rebuttal period has closed, and we understand that any rebuttals posted late are not to be formally considered. We are posting this information now simply because these longer experiments are computationally intensive and have only recently concluded. These results align perfectly with and amplify our earlier findings.
> >
> >
> > | Model Name           | Unambiguous (3k/5k)  | Ambiguous (3k/5k) |
> > | -------------------- | -------------------- | ----------------- |
> > | llama3-8B-Instruct   |   0.98/0.99   |  0.69/0.44  |
> > | llama3-70B-Instruct  |   0.99/0.99   |  0.94/0.76  |
> > | qwen2.5-7B-Instruct  |   0.99/0.99   |  0.43/0.19  |
> > | qwen2.5-32B-Instruct |   0.99/0.99   |  0.94/0.93  |
> >
> > Thank you again for your time. We remain eager to hear any questions you may have.

---

> > > ### Author Response · Authors · 2025-08-07
> > >
> > > We thank the reviewer once again for their insightful commens and feedback. As the discussion period is coming to the end, we would like to briefly summarize the key points from our response to the reviewer's concerns:
> > >
> > > 1. Scaling Lorem-Ipsum results up to 5.000 tokens confirms our findings: i.e there is a lower success rate for ambiguous bigrams than for unambiguous ones.
> > > 2. Comparison to from-scratch models is provided in Appendix B, showing that the asymmetry can be attributed to pre-training.
> > > 3. In our response, we explain why these tasks are fundamental to reasoning:  solving subproblems in the reasoning chain often involves saome form of retrieving and copying the input (reference [2] above).
> > > 4. We will make the writing more explicit, e.g. in regards to explicit takeways following each section.

---

### Note · Authors · 2025-08-12

We appreciate the engagement from the reviewers, and thank all those reviewers who interacted with us during the discussion period.
We are very pleased with the fact that we were able to address the concerns of all the reviewers, which also led to **reviewer gWjY increasing their score, for which we are very grateful**. We are also glad that we could engage with reviewers KxkG and akQM. To us, it seemed that reviewer KxkG's concerns were primarily regarding presentation and not methodology and we will strive to incorporate all suggestions made by them. We are also thankful to reviewer akQM for their positive assessment and actionable suggestions for improved presentation and clarity, which we will implement.

Unfortunately, we were not able to engage with Reviewer JBEt, who initially had a borderline assessment of our work. We would like to reiterate why we believe that their concerns have been addressed:
* First, JBEt was concerned about the short input lengths for our copying task. In our response, **we scaled the inputs lengths up to 5,000 tokens for the copying task (i.e., lorem-ipsum task). The theoretically predicted difference between deterministic vs non-deterministic bigrams was borne out throughout**.
* Second, JBEt was concerned about our argument that the left-right/forward-backward asymmetry is caused by pretraining. We pointed out that this is shown by the results in Appendix B, where we show that **models trained from scratch (no pretraining) don't have this asymmetry**.
* Third, JBEt asked about the overall relevance of our tasks and findings, which we addressed in the rebuttal.
* Fourth, JBEt suggested better storytelling; we are grateful for this suggestion and committed to implementing it.

Taken together, we thank all reviewers for the constructive and sensible feedback, and are happy to have addressed all concerns.

---

### Decision · Program_Chairs · 2025-09-17

**Decision:**

Accept (poster)

**Comment:**

The paper evaluates whether large-scale pretraining overcomes Transformers’ length generalization limits (using C-RASP) when evaluated on synthetic retrieval and copying tasks. It finds a consistent bias favoring forward induction over backward anti-induction, linked to circuit asymmetries shaped by pretraining. Fine-tuning restores symmetry, but core architectural constraints remain. The reviewers have expressed various concerns over the experiments, but also over the significance of the contribution. Given the general title, I do agree with the reviewers that I expected more and agree with the reviewers that there seems to be a wrong scope. Therefore, I will recommend acceptance since this is an interesting observation and analysis on inductive bias, but I **strongly recommend** the authors to revise the title and make it more descriptive of their work in the camera-ready version.